# List and Certificate Complexities in Replicable Learning

**Peter Dixon**[1]**, A. Pavan** [2]**, Jason Vander Woude**[3]**, and  N. V. Vinodchandran**[4]

[1]*Independent Researcher*
*tooplark@gmail.com*
[2]*Iowa State University*
*pavan@cs.iastate.edu*
[3]*University of Nebraska-Lincoln*
*jasonvw@huskers.unl.edu*
[4]*University of Nebraska-Lincoln*
*vinod@cse.unl.edu*

## Abstract

We investigate replicable learning algorithms. Informally a learning algorithm is replicable if the algorithm outputs the same canonical hypothesis over multiple runs with high probability, even when different runs observe a different set of samples from the unknown data distribution. In general, such a strong notion of replicability is not achievable. Thus we consider two feasible notions of replicability called *list replicability* and *certificate replicability*. Intuitively, these notions capture the degree of (non) replicability. The goal is to design learning algorithms with optimal list and certificate complexities while minimizing the sample complexity. Our contributions are the following.

– We first study the learning task of estimating the biases of $d$ coins, up to an additive error of $\varepsilon$, by observing samples. For this task, we design a $(d+1)$-list replicable algorithm. To complement this result, we establish that the list complexity is optimal, i.e there are no learning algorithms with a list size smaller than $d+1$ for this task. We also design learning algorithms with certificate complexity $\tilde{O}(\log d)$. The sample complexity of both these algorithms is $\tilde{O}(\frac{d^2}{\varepsilon^2})$ where $\varepsilon$ is the approximation error parameter (for a constant error probability).

– In the PAC model, we show that any hypothesis class that is learnable with $d$-nonadaptive statistical queries can be learned via a $(d+1)$-list replicable algorithm and also via a $\tilde{O}(\log d)$-certificate replicable algorithm. The sample complexity of both these algorithms is $\tilde{O}(\frac{d^2}{\nu^2})$ where $\nu$ is the approximation error of the statistical query. We also show that for the concept class $d$-THRESHOLD, the list complexity is exactly $d+1$ with respect to the uniform distribution.

To establish our upper bound results we use rounding schemes induced by geometric partitions with certain properties. We use Sperner/KKM Lemma to establish the lower bound results.

37th Conference on Neural Information Processing Systems (NeurIPS 2023).

# 1 Introduction

Replicability and reproducibility in science are critical concerns. The fundamental requirement that scientific results and experiments be replicable/reproducible is central to the development and evolution of science. In recent years, these concerns have grown as several scientific disciplines turn to data-driven research, which enables exponential progress through data democratization and affordable computing resources. The replicability issue has received attention from a wide spectrum of entities, from general media publications (for example, The Economist's "How Science Goes Wrong," 2013 (eco13)) to scientific publication venues (for example, see (JP05; Bak16)) to professional and scientific bodies such as the National Academy of Sciences, Engineering, and Medicine (NASEM). The emerging challenges to replicability and reproducibility have been discussed in depth by a consensus study report published by NASEM (NAS19).

A broad approach taken to ensure the reproducibility/replicability of algorithms is to make the datasets, algorithms, and code publicly available. Of late, conferences have been hosting replicability workshops to promote best practices and to encourage researchers to share code (see (PVLS$^+$21) and (MPK19)). An underlying assumption is that consistent results can be obtained using the same input data, computational methods, and code. However, these practices alone are insufficient to ensure replicability as modern-day approaches use computations that inherently involve randomness.

Computing over random variables results in a high degree of non-replicability, especially in machine learning tasks. Machine learning algorithms observe samples from a (sometimes unknown) distribution and output a hypothesis. Such algorithms are inherently non-replicable. Two distinct runs of the algorithm will output different hypotheses as the algorithms see different sets of samples over the two runs. Ideally, to achieve "perfect replicability," we would like to design algorithms that output the same canonical hypothesis over multiple runs, even when different runs observe a different set of samples from the unknown distribution.

We first observe that perfect replicability is not achievable in learning, as a dependency of the output on the data samples is inevitable. We illustrate this with a simple learning task of estimating the bias of a coin: given $n$ independent tosses of a coin with unknown bias $b$, output an estimate of $b$ that is within an additive error of $\varepsilon$ with high probability. It is relatively easy to argue that there is no algorithm that outputs a *canonical estimate* $v_b$ with probability $\geq 2/3$ so that $|v_b - b| \leq \varepsilon$. Suppose, for the sake of contradiction, such an algorithm $A$ exists. Consider a sequence of coins with biases $b_1 < b_2 < \cdots < b_m$ where each $b_{i+1} - b_i \leq \eta = 1/10n$, and $b_m - b_1 \geq 2\varepsilon$. For two adjacent biases $b_i$ and $b_{i+1}$, the statistical distance (denoted by $d_{\mathrm{TV}}$) between $\mathcal{D}_{i+1}^n$ and $\mathcal{D}_i^n$ is $\leq n\eta$, where $\mathcal{D}_i^n$ is the distribution of $n$ independent tosses of the $i^{th}$ coin. Let $v_{i+1}$ and $v_i$ be the canonical estimates output by the algorithm for biases $b_{i+1}$ and $b_i$ respectively. Since $A$ on samples from distribution $\mathcal{D}_i^n$ outputs $v_i$ with probability at least $2/3$ and $d_{\mathrm{TV}}(D_i^n, D_{i+1}^n) \leq n\eta$, $A(\mathcal{D}_{i+1}^n)$ must output $v_i$ with probability at least $2/3 - n\eta$. Since $A(\mathcal{D}_{i+1}^n)$ must output a canonical value $v_{i+1}$ with probability at least $2/3$, this implies that $v_i = v_{i+1}$ (if not the probabilities will add up to $> 1$). Thus, on all biases $b_1, \ldots, b_m$, the algorithm $A$ should output the same value. This leads to a contradiction since $b_1$ and $b_m$ are $2\varepsilon$ apart. However, it is easy to see that there is an algorithm for bias-estimation that outputs one of *two canonical* estimates with high probability using $n = O(1/\varepsilon^2)$ tosses: estimate the bias within an error of $\varepsilon/2$ and round the value to the closest multiple of $\varepsilon$. The starting point of our work is these two observations. Even though it may not be possible to design learning algorithms that are perfectly replicable, it is possible to design algorithms whose "non-replicability" is minimized.

We study two notions of replicability called *list replicability* and *certificate replicability* which quantify the *degree* of (non)-replicability of learning algorithms. They are rooted in the pseudodeterminism-literature (GG11; Gol19; GL19) which studied concepts known as *multi-pseudodeterminism* (related to list replicability) and *influential bit algorithms* (related to certificate replicability). Recently a notion of *algorithmic stability* and its variants such as *list global stability*, *pseudo global stability*, and *ρ-replicability* have been studied in the context of learning algorithms (BLM20; GKM21; ILPS22). These notions are very similar to each other and Section 2 discusses the similarities and connections.

## 1.1 Our Results

Informally, an algorithm $A$ is $k$-list replicable if there is a list $L$ consisting of $k$ (approximately correct) hypotheses so that the output of the algorithms $A$ belongs to $L$ with high probability. This implies that when the algorithm $A$ is run multiple times, we see at most $k$ distinct hypotheses (with

high probability). The value $k$ is called the *list complexity* of the algorithm. An algorithm whose list complexity is 1 is perfectly replicable. Thus, list complexity can be considered as a degree of (non) replicability. The goal in this setting is to design learning algorithms that minimize the list complexity $k$ for various learning tasks.

In certificate replicability, the learning algorithm has access to an $\ell$-bit random string that is independent of samples and the other randomness that the algorithm may use. It is required that for most $\ell$-bit random strings $r$, the algorithm must output a *canonical* (approximately correct) hypothesis $h_r$ that depends only on $r$. Thus once we fix a good $\ell$-bit random string $r$, multiple runs of the algorithm will output the same hypothesis with high probability. We call $\ell$ the *certificate complexity* of the algorithm. An algorithm with zero certificate complexity is perfectly replicable. Thus $\ell$ is another measure of the degree of (non) replicability of the algorithm. The goal in this setting is to design learning algorithms that minimize the certificate complexity.

A critical resource in machine learning tasks is the *sample complexity*—the number of samples that the algorithm observes. This work initiates a study of learning algorithms that are efficient in list complexity, certificate complexity as well as sample complexity. The main contribution of this work is the design of algorithms with an optimal list and certificate complexities with efficient sample complexity for a few fundamental learning tasks.

**Estimating the biases of $d$ coins.**   We consider the basic problem of estimating the biases of $d$ coins simultaneously by observing $n$ tosses of each of the coins which we call $d$-COIN BIAS ESTIMATION PROBLEM. The task is to output an approximate bias vector $\vec{v}$ with probability at least $(1 - \delta)$ so that $\|\vec{b} - \vec{v}\|_\infty \leq \varepsilon$ where $\vec{b} = \langle b_1, \cdots b_d \rangle$ is the true bias vector, i.e., $b_i$ is the true bias of the $i$th coin. For this task, we establish the following results.

- There is a $(d + 1)$-list replicable learning algorithm for $d$-COIN BIAS ESTIMATION PROBLEM whose sample complexity (number of observed coin tosses) is $n = O(\frac{d^2}{\varepsilon^2} \cdot \log \frac{d}{\delta})$ per coin.
- There is a $\lceil \log \frac{d}{\delta} \rceil$-certificate replicable algorithm for $d$-COIN BIAS ESTIMATION PROBLEM with sample complexity $n = O(\frac{d^2}{\varepsilon^2 \delta^2})$ per coin.
- We establish the optimality of the above upper bounds in terms of list complexity. We show that there is no $d$-list replicable learning algorithm for $d$-COIN BIAS ESTIMATION PROBLEM.

While $d$-COIN BIAS ESTIMATION PROBLEM is a basic learning task and is of interest by itself, the techniques developed for this problem are applicable in the context of PAC learning.

**PAC learning.**   We investigate list and certificate replicable PAC learning algorithms.

- We establish the following generic result: Any concept class that can be learned using $d$ non-adaptive statistical queries can be learned by a $(d + 1)$-list replicable PAC learning algorithm with sample complexity $O(\frac{d^2}{\nu^2} \cdot \log \frac{d}{\delta})$ where $\nu$ is the statistical query parameter. We also show that such concept classes admit a $\lceil \log \frac{d}{\delta} \rceil$-certificate replicable PAC learning algorithm with sample complexity $O(\frac{d^2}{\nu^2 \delta^2} \cdot \log \frac{d}{\delta})$.
- We study the list complexity of the concept class $d$-THRESHOLD. Each hypothesis $h_{\vec{t}}$ in this concept class is described via a $d$-dimensional vector $\vec{t} \in [0, 1]^d$. For a hypothesis $h_t$, $h_t(\vec{x}) = 1$ if and only if $x_i \leq t_i$, $1 \leq i \leq d$[1]. We establish that, under the uniform distribution, the list complexity of $d$-THRESHOLD is *exactly* $d + 1$.

To establish our upper bound results we use rounding schemes induced by geometric partitions with certain properties. We use Sperner/KKM Lemma as a tool to establish the lower bound results. Due to space restrictions, most of the proofs are given in the supplementary material.

## 2   Prior and Related Work

Formalizing reproducibility and replicability has gained considerable momentum in recent years. While the terms reproducibility and replicability are very close and often used interchangeably, there

---

[1]This concept class is the same as the class of axis-parallel rectangles in $[0, 1]^d$ with $\vec{0}$ as one of the corners.

has been an effort to distinguish between them and accordingly, our notions fall in the replicability definition (PVLS⁺21). As mentioned earlier, reproducibility, replicability, and related notions have been of considerable interest in the context of traditional randomized algorithms and learning algorithms. Here we discuss the prior work that is most relevant to the present work. A more detailed discussion of the prior and related work is provided in the supplementary material.

The seminal work of (BLM20) defined the notion of *global stability* in the context of learning, which is similar to the notion of pseudodeterminism (GG11) in the context of traditional algorithms. They define a learning algorithm $A$ to be $(n, \eta)$-globally stable with respect to a distribution $D$ if there is a hypothesis $h$ such that $\Pr_{S \sim D^n}(A(S) = h) \geq \eta$, here $\eta$ is called the *stability parameter*. Using this notion as an intermediate tool they established that every concept class with finite Littlestone dimension can be learned by an approximate differentially private algorithm. The work in (GKM21) extended the notion of global stability to *list-global stability* and *pseudo-global stability*. The authors of (GKM21) used these concepts to design user-level differentially private algorithms. The notion of pseudo-global stability is similar to the notion of certificate applicability, however, as defined, the notion of list-global stability differs from our notion of list replicability.

The recent work reported in (ILPS22) introduced the notion of $\rho$-*replicability*. A learning algorithm $A$ is $\rho$-replicable if $\Pr[A(S_1, r) = A(S_2, r)] \geq 1 - \rho$, where $S_1$ and $S_2$ are samples drawn from a distribution $\mathcal{D}$ and $r$ is the internal randomness of the learning algorithm $A$. They designed replicable algorithms for many learning tasks, including statistical queries, approximate heavy hitters, median, and learning half-spaces. It is known that the notions of pseudo-global stability and $\rho$-replicability are the same up to polynomial factors in the parameters (ILPS22; GKM21).

The present work introduces the notions of list and certificate complexities as measures of the degree of (non) replicability. Our goal is to design learning algorithms with optimal list and certificate complexities while minimizing the sample complexity. The earlier works did not focus on minimizing these quantities. The works of (BLM20; GKM21) used replicable algorithms as an intermediate step to design differentially private algorithms. The work of (ILPS22) did not consider reducing the certificate complexity in their algorithms and also did not study list replicability. The main distinguishing feature of our work from prior works is our focus on designing learning algorithms that are efficient in list, certificate, and sample complexities as well as establishing optimality results for list and certificate complexity.

A very recent and independent work of (CMY23) investigated relations between list replicability and the stability parameter $\nu$, in the context of distribution-free PAC learning. They showed that for every concept class $\mathcal{H}$, its list complexity is exactly the inverse of the stability parameter. They also showed that the list complexity of a hypothesis class is at least its VC dimension. For establishing this they exhibited, for any $d$, a concept class whose list complexity is exactly $d$. There are some similarities between their work and the present work. We establish similar upper and lower bounds on the list complexity but for different learning tasks: $d$-THRESHOLD and $d$-COIN BIAS ESTIMATION PROBLEM. For $d$-THRESHOLD, our results are for PAC learning under *uniform* distribution and do not follow from their distribution-independent results. Thus our results, though similar in spirit, are incomparable to theirs. Moreover, their work did not focus on efficiency in sample complexity and also did not study certificate complexity which is a focus of our paper. We do not study the stability parameter.

Finally, we would like to point out that there notions of list PAC learning and list-decodable learning in the learning theory literature (see (CP23) and (RY20) for recent progress on these notions). However, these notions are different from the list replicable learning that we consider in this paper. List PAC learning and list-decodable learning are generalized models of PAC learning. For example, any learning task that is PAC learnable is trivially list PAC learnable with a list size of 1. However, list replicable learning is an additional requirement that needs to be satisfied by a learner. Thus the notion of list and list-decodable PAC learning are different from the notions of list/certificate replicability.

## 3   List and Certificate Replicability Notions

We define list and certificate replicability notions for general learning tasks. A learning problem is a family $\mathcal{D}$ of distributions over a domain $\mathcal{X}$, a set $\mathcal{H}$ (representing hypotheses) and an error function $err : \mathcal{D} \times \mathcal{H} \to [0, \infty)$. The goal is to learn a hypothesis from $\mathcal{H}$ by observing samples from a distribution $D$ where $D \in \mathcal{D}$ with a small error $err(D, h)$. A learning algorithm $A$ has the following

inputs: (i) $m$ independent samples from a distribution $D \in \mathcal{D}$ and (ii) $\varepsilon \in (0, \infty)$ and (iii) $\delta \in (0, 1]$. It may also receive additional inputs.

**Definition 3.1** (List Replicability). Let $k \in \mathbb{N}$, $\varepsilon \in (0, \infty)$, and $\delta \in [0, 1]$. A learning algorithm $A$ is called $(k, \varepsilon, \delta)$-list replicable if the following holds: There exists $n \in \mathbb{N}$ such that for every $D \in \mathcal{D}$, there exists a list $L \subseteq \mathcal{H}$ of size at most $k$ such that (i) for all $h \in L$, $err(D, h) \leq \varepsilon$, and (ii)

$$\Pr_{s \sim D^n} [A(s, \varepsilon, \delta) \in L] \geq 1 - \delta.$$

For $k \in \mathbb{N}$, we call $A$ *k-list replicable* if for all $\varepsilon \in (0, \infty)$ and $\delta \in (0, 1]$, $A$ is $(k, \varepsilon, \delta)$-list replicable. We say that $n$ is the *sample complexity* of $A$ and $k$ is the *list complexity* of $A$.

**Definition 3.2** (Certificate Replicability). Let $\ell \in \mathbb{N}$, $\varepsilon \in (0, \infty)$, and $\delta \in [0, 1]$. A learning algorithm $A$ is called $(\ell, \varepsilon, \delta)$-certificate replicable if the following holds: There exists $n \in \mathbb{N}$ such that for every $D \in \mathcal{D}$ there exists $h : \{0, 1\}^\ell \to \mathcal{H}$ such that

$$\Pr_{r \in \{0,1\}^\ell} \left[ \Pr_{s \in D^n} \left[ A(s, \varepsilon, \delta, r) = h(r) \text{ and } err(D, h(r)) \leq \varepsilon \right] \geq 1 - \delta \right] \geq 1 - \delta.$$

We can refine the above definition by introducing another probability parameter $\rho$ and define a notion of $(\ell, \varepsilon, \rho, \delta)$-replicability. The definition is the same as above except that we require the outer probability to be $1 - \rho$, and the inner probability is $1 - \delta$. For simplicity, we take $\rho$ to be $\delta$ and work with the notion of $(\ell, \varepsilon, \delta)$-replicability.

The above definitions generalize the notion of *perfect replicability*, where it is required that the learning algorithm outputs a canonical hypothesis $h$ (with $err(D, h) \leq \varepsilon$) with probability $\geq 1 - \delta$. A motivation for these definitions is to characterize how close we can be to perfect replicability in scenarios where if perfect replicability is not achievable. Note that for list (certificate) replicability, when $k = 1$ (respectively, $\ell = 0$), we achieve perfect applicability. We note that the above definitions are inspired by multi-pseudodeterminism (Gol19) and influential-bit algorithms (GL19).

## 4  Primary Lemmas

Our upper bounds are based on certain rounding schemes and the lower bound is based on Sperner/KKM lemma. In this section, we state a few key technical lemmas that will be used in the rest of the paper. The proofs are in the supplementary material. We will use the following notation. We use $\mathrm{diam}_\infty$ to indicate the diameter of a set relative to the $\ell_\infty$ norm and $\overline{B}_\varepsilon^\infty(\vec{p})$ to represent the closed ball of radius $\varepsilon$ centered at $\vec{p}$ relative to the $\ell_\infty$ norm. That is, in $\mathbb{R}^d$ we have $\overline{B}_\varepsilon^\infty(\vec{p}) = \prod_{i=1}^d [p_i - \varepsilon, p_i + \varepsilon]$.

We first state a lemma that gives a universal deterministic rounding algorithm that is used in designing list replicable algorithms. The lemma is based on the work in (VWDP$^+$22) and is a byproduct of certain geometric partitions they call *secluded partitions*.

**Lemma 4.1.** Let $d \in \mathbb{N}$ and $\varepsilon \in (0, \infty)$. Let $\varepsilon_0 = \frac{\varepsilon}{2d}$. There is an efficiently computable function $f_\varepsilon : \mathbb{R}^d \to \mathbb{R}^d$ with the following two properties:

1. *For any $x \in \mathbb{R}^d$ and any $\hat{x} \in \overline{B}_{\varepsilon_0}^\infty(x)$ it holds that $f_\varepsilon(\hat{x}) \in \overline{B}_\varepsilon^\infty(x)$.*

2. *For any $x \in \mathbb{R}^d$ the set $\left\{ f_\varepsilon(\hat{x}) \colon \hat{x} \in \overline{B}_{\varepsilon_0}^\infty(x) \right\}$ has cardinality at most $d + 1$.*

Item (1) states that if $\hat{x}$ is an $\varepsilon_0$ approximation of $x$, then $f_\varepsilon(\hat{x})$ is an $\varepsilon$ approximation of $x$, and Item (2) states that $f_\varepsilon$ maps every $\varepsilon_0$ approximation of $x$ to one of at most $d + 1$ possible values.

The following lemma gives a universal randomized rounding algorithm that is used in designing certificate replicable algorithms. We note that randomized rounding schemes have been used in a few prior works (SZ99; DPV18; Gol19; GL19; ILPS22). Our rounding scheme is more nuanced as it is geared towards minimizing certificate complexity.

**Lemma 4.2.** Let $d \in \mathbb{N}$, $\varepsilon_0 \in (0, \infty)$ and $0 < \delta < 1$. There is an efficiently computable deterministic function $f : \{0, 1\}^\ell \times \mathbb{R}^d \to \mathbb{R}^d$ with the following property. For any $x \in \mathbb{R}^d$,

$$\Pr_{r \in \{0,1\}^\ell} \left[ \exists x^* \in \overline{B}_\varepsilon^\infty(x) \ \forall \hat{x} \in \overline{B}_{\varepsilon_0}^\infty(x) : f(r, \hat{x}) = x^* \right] \geq 1 - \delta$$

where $\ell = \lceil \log \frac{d}{\delta} \rceil$ and $\varepsilon = (2^\ell + 1)\varepsilon_0 \leq \frac{2\varepsilon_0 d}{\delta}$.

The following result is a corollary to a cubical variant of Sperner's lemma/KKM lemma initially developed in (DLPES02) and expanded on in (VWDP+22). We use this to establish our lower bound results.

**Lemma 4.3.** *Let $\mathcal{P}$ be a partition of $[0,1]^d$ such that for each member $X \in \mathcal{P}$, it holds that $\mathrm{diam}_\infty(X) < 1$. Then there exists $\vec{p} \in [0,1]^d$ such that for all $\delta > 0$ we have that $\overline{B}_\delta^\infty(\vec{p})$ intersects at least $d+1$ members of $\mathcal{P}$.*

## 5 Replicability of Learning Coins Biases

In this section, we establish replicability results for estimating biases of $d$ coins.

**Definition 5.1.** The $d$-COIN BIAS ESTIMATION PROBLEM is the following problem: Design an algorithm $A$ (possibly randomized) that gets $\varepsilon \in (0,1)$, $\delta \in (0,1]$ as inputs, observes independent tosses of an ordered collection of $d$-many biased coins with a bias vector $\vec{b} \in [0,1]^d$, and outputs $\vec{v}$ so that $\|\vec{b} - \vec{v}\|_\infty \le \varepsilon$ with probability $\ge 1 - \delta$.

The $d$-COIN BIAS ESTIMATION PROBLEM fits in the framework of the general learning task introduced in Section 3 so that we can talk about list and certificate replicable algorithm for $d$-COIN BIAS ESTIMATION PROBLEM. We describe this now.

For $d$-COIN BIAS ESTIMATION PROBLEM we have the following. $\mathcal{X} = \{0,1\}^d$ (where 0 corresponds to Tail and 1 corresponds to Head) which is the set of representations of all possibilities of flipping $d$-many coins. The class of distributions $\mathcal{D}$ is the set of all $d$-fold products of Bernoulli distributions. Each distribution $D \in \mathcal{D}$ is parameterized with a vector $v_D = \langle b_1, \cdots, b_d \rangle \in [0,1]^d$. When sampled according to $v_D$ we obtain a sample point $\langle x_1 \cdots x_d \rangle \in \mathcal{X}$, where $\Pr(x_i = 1) = b_i$. We take the class of hypothesis $\mathcal{H}$ to be $[0,1]^d$ which is the set of all $d$-tuples representing biases of a collection of $d$-many coins. Lastly, the error function is defined as $err(D, h) = \|v_D - h\|_\infty$.

### 5.1 Replicable Algorithms for $d$-COIN BIAS ESTIMATION PROBLEM

In this section, we design list and certificate replicable algorithms for $d$-COIN BIAS ESTIMATION PROBLEM .

**Theorem 5.2.** *There exists an $(d+1)$-list replicable algorithm for $d$-COIN BIAS ESTIMATION PROBLEM. For a given $\varepsilon$ and $\delta$, its sample complexity is $n = O(\frac{d^2}{\varepsilon^2} \cdot \log \frac{d}{\delta})$, per coin.*

---

**Algorithm 1** $(d+1)$-list replicable algorithm for $d$-COIN BIAS ESTIMATION PROBLEM

---

**Input:** $\varepsilon > 0, \delta \in (0,1]$, sample access to $d$ coins with biases $\vec{b} \in [0,1]^d$

$\varepsilon_0 \stackrel{\text{def}}{=} \frac{\varepsilon}{2d}, \delta_0 \stackrel{\text{def}}{=} \frac{\delta}{d}$

$n \stackrel{\text{def}}{=} O\left(\frac{\ln(1/\delta_0)}{\varepsilon_0^2}\right) = O\left(\frac{d^2 \ln(d/\delta)}{\varepsilon^2}\right)$ for some constant

Let $f_\varepsilon : \mathbb{R}^d \to \mathbb{R}^d$ be as in Lemma 4.1.

Let $g : \mathbb{R}^d \to [0,1]^d$ be the function which restricts coordinates to the unit interval (i.e.

$$g(\vec{y}) \stackrel{\text{def}}{=} \left\langle \begin{cases} 0 & y_i < 0 \\ y_i & y_i \in [0,1] \\ 1 & y_i > 1 \end{cases} \right\rangle_{i=1}^d$$

)

Take $n$ samples from each coin and let $\vec{a}$ be the empirical biases.

**return** $g(f(\vec{a}))$

---

*Proof.* Note that when $\varepsilon \ge 1/2$, a trivial algorithm that outputs a vector with $1/2$ in each component works. Thus the most interesting case is when $\varepsilon < 1/2$. Our list replicable algorithm is described in Algorithm 1. We will prove its correctness by associating for each possible bias $\vec{b} \in [0,1]^d$, a set $L_{\vec{b}}$ with the three necessary properties: (1) $|L_{\vec{b}}| \le d+1$, (2) $L_{\vec{b}} \subseteq \overline{B}_\varepsilon^\infty(\vec{b})$ (and also the problem specific

restriction that $L_{\vec{b}} \subseteq [0,1]^d$), and (3) when given access to coins of biases $\vec{b}$, with probability at least $1 - \delta$ the algorithm returns a value in $L_{\vec{b}}$.

Let $L_{\vec{b}} = \left\{ g(f_\varepsilon(\vec{x})) \colon \vec{x} \in \overline{B}^\infty_{\varepsilon_0}(\vec{b}) \right\}$. By Lemma 4.1, $f_\varepsilon$ takes on at most $d + 1$ values on $\overline{B}^\infty_{\varepsilon_0}(\vec{b})$ (which means $g \circ f_\varepsilon$ also takes on at most $d+1$ values on this ball) which proves that $|L_{\vec{b}}| \leq d + 1$. This proves property (1).

Next we state the following observation which says that the coordinate restriction function $g$ of Algorithm 1 does not reduce approximation quality. The proof is straightforward.

**Observation 5.3.** *Using the notation of Algorithm 1, if $\vec{y} \in \overline{B}^\infty_\varepsilon(\vec{b})$ then $g(\vec{y}) \in \overline{B}^\infty_\varepsilon(\vec{b})$.*

We now establish Property (2). We know from Lemma 4.1 that for each $\vec{x} \in \overline{B}^\infty_{\varepsilon_0}(\vec{b})$ we have $f_\varepsilon(\vec{x}) \in \overline{B}^\infty_\varepsilon(\vec{b})$, and by Observation 5.3, $g$ maintains this quality and we have $g(f_\varepsilon(\vec{x})) \in \overline{B}^\infty_\varepsilon(\vec{b})$. This shows that $L_{\vec{b}} \subseteq \overline{B}^\infty_\varepsilon(\vec{b})$ proving property (2).

By Chernoff's bounds, for a single biased coin, with $n = O\left( \frac{\ln(1/\delta_0)}{\varepsilon_0^2} \right)$ independent samples of the coin we can estimate the bias within $\varepsilon_0$ with probability at least $1 - \delta_0$. Thus, by a union bound, if we take $n$ samples of each of the $d$ coins, there is a probability of at most $d \cdot \delta_0 = \delta$ that at least one of the empirical coin biases is not within $\varepsilon_0$ of the true bias. Thus, by taking $n$ samples of each coin, we have with probability at least $1 - \delta$ that the empirical biases $\vec{a}$ belong to $\overline{B}^\infty_{\varepsilon_0}(\vec{b})$. In the case that this occurs, we have by definition of $L_{\vec{b}}$ that the value $g(f_\varepsilon(\vec{a}))$ returned by the algorithm belongs to the set $L_{\vec{b}}$. This proves property (3). The sample complexity follows since $\varepsilon_0 = \frac{\varepsilon}{2d}$ and $\delta_0 = \frac{\delta}{d}$. $\qquad\square$

The next result is on certificate replicable algorithm for $d$-COIN BIAS ESTIMATION PROBLEM.

**Theorem 5.4.** *For every $\varepsilon$ and $\delta$, there is a $(\lceil \log \frac{d}{\delta} \rceil, \varepsilon, \delta)$-certificate replicable algorithm for $d$-COIN BIAS ESTIMATION PROBLEM with sample complexity of $n = O(\frac{d^2}{\varepsilon^2 \delta^2})$ per coin.*

*Proof.* Let $\varepsilon$ and $\delta$ be the input parameters to the algorithm and $\vec{b}$ the bias vector. Set $\varepsilon_0 = \frac{\varepsilon \delta}{2d}$. The algorithm $A$ first estimates the bias of each coin with up to $\varepsilon_0$ with a probability error parameter $\frac{\delta}{d}$ using a standard estimation algorithm. Note that this can be done using $O(\frac{d^2}{\varepsilon^2 \delta^2})$ tosses per coin. Let $\vec{v}$ be the output vector. It follows that $\vec{v} \in \overline{B}^\infty_{\varepsilon_0}(\vec{b})$ with probability at least $1 - \delta$. Then it runs the deterministic function $f$ described in Lemma 4.2 with input $r \in \{0,1\}^\ell$ with $\ell = \lceil \log \frac{d}{\delta} \rceil$ and $\vec{v}$ and outputs the value of the function. Lemma 4.2 guarantees that for $1 - \delta$ fraction of the $r$s, all $\vec{v} \in \overline{B}^\infty_{\varepsilon_0}(\vec{b})$ gets rounded to the same value by $f$. Hence algorithm $A$ satisfies the requirements of the certificate-replicability. The certificate complexity is $\lceil \log \frac{d}{\delta} \rceil$. $\qquad\square$

We remark that by using the refined definition of certificate replicability mentioned in Section 3, we can obtain a $(\lceil \log \frac{d}{\rho} \rceil, \varepsilon, \rho, \delta)$-replicable algorithm with sample complexity $O(\frac{d^2}{\varepsilon^2 \rho^2} \log(1/\delta))$. Note that an $\ell$-certificate replicable algorithm leads to a $2^\ell$-list replicable algorithm. Thus Theorem 5.4 gives a $O(\frac{d}{\delta})$-list replicable algorithm for $d$-COIN BIAS ESTIMATION PROBLEM with sample complexity $O(\frac{d^2}{\varepsilon^2 \delta^2})$. However, this is clearly sub-optimal and Theorem 5.2 gives an algorithm with a much smaller list and sample complexities. Also from the work of Goldrecih (Gol19), it follows that $\ell$-list replicable algorithm can be converted into a $\lceil \log(\frac{\ell}{\delta}) \rceil$-certificate replicable algorithm. However, this conversion increases the sample complexity. For example, when applied to $d$-COIN BIAS ESTIMATION PROBLEM, the sample complexity becomes $O(\frac{d^6}{\varepsilon^2 \delta^2})$. In comparison, Theorem 5.4 uses a tailored rounding technique to achieve an algorithm with a much smaller sample complexity.

## 5.2 An Impossibility Result

This section establishes the optimality of the list complexity for $d$-COIN BIAS ESTIMATION PROBLEM.

**Theorem 5.5.** *For $k < d + 1$, there does not exist a $k$-list replicable algorithm for the $d$-COIN BIAS ESTIMATION PROBLEM.*

Before proving the theorem, we need a lemma that follows from the Data Processing Inequality; for more detail see the supplementary material. In the following, we use $\mathcal{D}_{A,\vec{b},n}$ to denote the distribution of the output of an algorithm for $d$-COIN BIAS ESTIMATION PROBLEM.

**Lemma 5.6.** *For biases $\vec{a}, \vec{b} \in [0,1]^d$ we have $d_{\mathrm{TV}}\left(\mathcal{D}_{A,\vec{a},n}, \mathcal{D}_{A,\vec{b},n}\right) \leq n \cdot d \cdot \|\vec{b} - \vec{a}\|_\infty.$*

*Proof.* We use the basic fact that an algorithm (deterministic or randomized) cannot increase the total variation distance between two input distributions.

The distribution giving one sample flip of each coin in a collection with bias $\vec{b}$ is the $d$-fold product of Bernoulli distributions $\prod_{i=1}^d \mathrm{Bern}(b_i)$ (which for notational brevity we denote as $\mathrm{Bern}(\vec{b})$, so the distribution which gives $n$ independent flips of each coin is the $n$-fold product of this and is denoted as $\mathrm{Bern}(\vec{b})^{\otimes n}$). We will show that for two bias vectors $\vec{a}$ and $\vec{b}$, $d_{\mathrm{TV}}\left(\mathrm{Bern}(\vec{b})^{\otimes n}, \mathrm{Bern}(\vec{a})^{\otimes n}\right) \leq n \cdot d \cdot \|\vec{b} - \vec{a}\|_\infty$. This suffices to establish the lemma.

Observe that we have for each $i \in [d]$,
$$d_{\mathrm{TV}}\left(\mathrm{Bern}(b_i), \mathrm{Bern}(a_i)\right) = |b_i - a_i|.$$

Hence we have
$$d_{\mathrm{TV}}\left(\mathrm{Bern}(\vec{b}), \mathrm{Bern}(\vec{a})\right) \leq \sum_{i=1}^d |b_i - a_i| \leq d \cdot \|\vec{b} - \vec{a}\|_\infty$$

and
$$d_{\mathrm{TV}}\left(\mathrm{Bern}(\vec{b})^{\otimes n}, \mathrm{Bern}(\vec{a})^{\otimes n}\right) \leq n \cdot d \cdot \|\vec{b} - \vec{a}\|_\infty.$$

$\square$

*Proof of Theorem 5.5.* Fix any $d \in \mathbb{N}$, and choose $\varepsilon$ and $\delta$ as $\varepsilon < \frac{1}{2}$ and $\delta \leq \frac{1}{d+2}$. Suppose for contradiction that such an algorithm $A$ does exists for some $k < d+1$. This means that for each possible bias vector $\vec{b} \in [0,1]^d$, there exists some set $L_{\vec{b}} \subseteq \mathcal{H}$ of hypotheses with three properties: (1) each element of $L_{\vec{b}}$ is an $\varepsilon$-approximation to $h_{\vec{b}}$, (2) $|L_{\vec{b}}| \leq k$, and (3) with probability at least $1 - \delta$, $A$ returns an element of $L_{\vec{b}}$.

By an averaging argument, this means that there exists at least one element in $L_{\vec{b}}$ which is returned by $A$ with probability at least $\frac{1}{k} \cdot (1 - \delta) \geq \frac{1}{k} \cdot (1 - \frac{1}{d+2}) = \frac{1}{k} \cdot \frac{d+1}{d+2} \geq \frac{1}{k} \cdot \frac{k+1}{k+2}$. Let $f \colon [0,1]^d \to [0,1]^d$ be a function which maps each bias $\vec{b}$ to such an element of $L_{\vec{b}}$. Since $\frac{1}{k} \cdot \frac{k+1}{k+2} > \frac{1}{k+1}$, let $\eta$ be such that $0 < \eta < \frac{1}{k} \cdot \frac{k+1}{k+2} - \frac{1}{k+1}$.

The function $f$ induces a partition $\mathcal{P}$ of $[0,1]^d$ where the members of $\mathcal{P}$ are the fibers of $f$ (i.e. $\mathcal{P} = \{f^{-1}(\vec{y}) \colon \vec{y} \in \mathrm{range}(f)\}$). By definition, for any member $X \in \mathcal{P}$ there exists some $\vec{y} \in \mathrm{range}(f)$ such that for all $\vec{b} \in X$, $f(\vec{b}) = \vec{y}$. By definition, we have $f(\vec{b}) \in L_{\vec{b}} \subseteq \overline{B}_\varepsilon^\infty(\vec{b})$ showing that $\vec{y} \in \overline{B}_\varepsilon^\infty(\vec{b})$ and by symmetry $\vec{b} \in \overline{B}_\varepsilon^\infty(\vec{y})$. This shows that $X \subseteq \overline{B}_\varepsilon^\infty(\vec{y})$, so $\mathrm{diam}_\infty(X) \leq 2\varepsilon < 1$.

Let $r = \frac{\eta \cdot d}{n}$. Since every member of $\mathcal{P}$ has $\ell_\infty$ diameter less than 1, by Lemma 4.3 there exists a point $\vec{p} \in [0,1]^d$ such that $\overline{B}_r^\infty(\vec{p})$ intersects at least $d+1 > k$ members of $\mathcal{P}$. Let $\vec{b}^{(1)}, \ldots, \vec{b}^{(d+1)}$ be points belonging to distinct members of $\mathcal{P}$ that all belong to $\overline{B}_r^\infty(\vec{p})$. By definition of $\mathcal{P}$, this means for distinct $j, j' \in [d+1]$ that $f(\vec{b}^{(j)}) \neq f(\vec{b}^{(j')})$.

Now, for each $j \in [d+1]$, because $\|\vec{p} - \vec{b}^{(j)}\|_\infty \leq r$, by Lemma 5.6 we have $d_{\mathrm{TV}}(\mathcal{D}_{A,\vec{p},n}, \mathcal{D}_{A,\vec{b}^{(j)},n}) \leq n \cdot d \cdot r = \eta$. However, this gives rise to a contradiction because the probability that $A$ with access to biased coins $\vec{b}^{(j)}$ returns $f(\vec{b}^{(j)})$ is at least $\frac{1}{k} \cdot \frac{k+1}{k+2}$, and by Lemma 5.6, it must be that $A$ with access to biased coins $\vec{p}$ returns $f(\vec{b}^{(j)})$ with probability at least $\frac{1}{k} \cdot \frac{k+1}{k+2} - \eta > \frac{1}{k+1}$; notationally, $\mathrm{Pr}_{\mathcal{D}_{A,\vec{b}^{(j)},n}}\left(\left\{f(\vec{b}^{(j)})\right\}\right) \geq \frac{1}{k} \cdot \frac{k+1}{k+2}$ and $d_{\mathrm{TV}}(\mathcal{D}_{A,\vec{b}^{(j)},n}, \mathcal{D}_{A,\vec{p},n}) \leq \eta$, so $\mathrm{Pr}_{\mathcal{D}_{A,\vec{p},n}}\left(\left\{f(\vec{b}^{(j)})\right\}\right) \geq \frac{1}{k} \cdot \frac{k+1}{k+2} - \eta > \frac{1}{k+1}$. This is a contradiction because a distribution cannot have $d+1 \geq k+1$ disjoint events that each have probability greater than $\frac{1}{k+1}$. $\square$

# 6 List and Certificate Replicability in PAC Learning

In this section, we establish replicability results for the PAC model. We first note that PAC model fits in the general learning framework introduced in Section 3. Consider the PAC learning model where we have a domain $\mathcal{X}'$, a concept class $\mathcal{C}$ (a family of subsets of $\mathcal{X}'$), and a hypothesis class $\mathcal{H}$ (a collection of functions $\mathcal{X}' \to \{0, 1\}$). For any distribution $D$ over $\mathcal{X}'$ and any concept indicator $f$ from $\mathcal{C}$, let $D_f$ denote the distribution over $\mathcal{X}' \times \{0, 1\}^d$ obtained by sampling $x \sim D$ and outputting $\langle x, f(x) \rangle$. We define the following learning problem in the general learning framework: $\mathcal{X} = \mathcal{X}' \times \{0, 1\}$, $\mathcal{H} = \mathcal{H}$, $\mathcal{D} = \{D_f : f \in \mathcal{C} \text{ and } D \text{ a distribution over } \mathcal{X}'\}$, and $err(D_f, h) = \Pr_{x \sim \mathcal{D}'}[f(x) \neq h(x)]$.

We are interested in statistical query learning which is defined by Kearns (Kea98).

**Definition 6.1.** A statistical query oracle $STAT(D_f, \nu)$ takes as an input a real-valued function $\phi : X \times \{0, 1\} \to (0, 1)$ and returns an estimate $v$ such that $|v - E_{\langle x, y \rangle \in D_f}[\phi(x, y)]| \leq \nu$ (Kea98). We say that an algorithm $A$ learns a concept class $\mathcal{H}$ via statistical queries if for every distribution $D$ and every function $f \in \mathcal{H}$, for every $0 < \varepsilon < 1$, there exists $\nu$ such that the algorithm $A$ on input $\varepsilon$, and $STAT(D_f, \nu)$ as an oracle, outputs a hypothesis $h$ such that $err(D_f, h) \leq \varepsilon$. The concept class is *non-adaptively* learnable if all the queries made by $A$ to the statistical query oracle are non-adaptive.

## 6.1 Replicable PAC Learning Algorithms

Next we state main results of this section. Due to space limitation, all the proofs are in the supplementary material. We first show that any concept class that is learnable with $d$ non-adpative statistical queries has a $(d + 1)$-list replicable algorithm.

**Theorem 6.2.** *Let $\mathcal{H}$ be a concept class that is learnable with $d$ non-adaptive statistical queries, then $\mathcal{H}$ is $(d + 1)$-list replicably learnable. Furthermore, the sample complexity $n = n(\nu, \delta)$ of the $(d + 1)$-list replicable algorithm is $O(\frac{d^2}{\nu^2} \cdot \log \frac{d}{\delta})$, where $\nu$ is the approximation error parameter of each statistical query oracle.*

We note that we can simulate a statistical query algorithm that makes $d$ *adaptive* queries to get a $2^d$-list replicable learning algorithm. This can be done by rounding each query to two possible values (the approximation factor increases by 2). The sample complexity of this algorithm will be $O(\frac{d}{\nu^2} \cdot \log \frac{d}{\delta})$. Next, we design a certificate replicable algorithm for hypothesis classes that admit statistical query learning algorithms.

**Theorem 6.3.** *Let $\mathcal{H}$ be a concept class that is learnable with $d$ non-adaptive statistical queries, then $\mathcal{H}$ is $\lceil \log \frac{d}{\delta} \rceil$-certificate replicably learnable. Furthermore, the sample complexity $n = n(\nu, \delta)$ of this algorithm is $O(\frac{d^2}{\nu^2 \delta^2} \cdot \log \frac{d}{\delta})$, where $\nu$ is the approximation error parameter of each statistical query oracle.*

We can also consider the case when the statistical query algorithm makes $d$ adaptive queries. In this case we get the following theorem. Note that the certificate complexity is close to linear in $d$ as oppose to logarithmic in the case of non-adaptive queries.

**Theorem 6.4.** *Let $\mathcal{H}$ be a concept class that is learnable with $d$ adaptive statistical queries, then $\mathcal{H}$ is $\lceil d \log \frac{d}{\delta} \rceil$-certificate replicably learnable. Furthermore, the sample complexity of this algorithm is $O(\frac{d^3}{\nu^2 \delta^2} \cdot \log \frac{d}{\delta})$, where $\nu$ is the approximation error parameter of each statistical query oracle.*

## 6.2 Impossibility Results in the PAC Model

In this section, we establish matching upper and lower bounds on the list complexity for the concept class $d$-THRESHOLD in the PAC model with respect to the uniform distribution. In particular, we establish that this learning task admits a $(d + 1)$-list replicable algorithm and does not admit a $d$-list replicable algorithm.

**Definition 6.5** ($d$-THRESHOLD). Fix some $d \in \mathbb{N}$. Let $X = [0, 1]^d$. For each value $\vec{t} \in [0, 1]^d$, let $h_{\vec{t}} : X \to \{0, 1\}$ be the concept defined as follows: $h_{\vec{t}}(\vec{x}) = 1$ if for every $i \in [d]$ it holds that $x_i \leq t_i$ and 0 otherwise. Let $\mathcal{H}$ be the hypothesis class consisting of all such threshold concepts: $\mathcal{H} = \{h_{\vec{t}} \mid \vec{t} \in [0, 1]^d\}$.

We first observe the impossibility of list-replicable algorithms in the distribution-free PAC model. This follows from known results.

**Observation 6.6.** *There is no $k$-list replicable algorithm (for any $k$) for $d$-THRESHOLD in the PAC model even when $d = 1$.*

The above observation follows from the works of (ALMM19) and (BLM20). It is known that $d$-THRESHOLD has an infinite Littlestone dimension. Suppose it admits $k$-list replicable algorithms in the PAC model. This implies that $d$-THRESHOLD is globally stable learnable with stability parameter $1/k$ (Please see (BLM20) for the definition of global stability). The work of (BLM20) showed that any class that is globally stable learnable with a constant stability parameter is differentially private learnable. The work of (ALMM19) showed that if a concept class is differentially private learnable, then it has a finite Littlestone dimension. Putting these results together, we obtain that if $d$-THRESHOLD admits $k$-list replicable algorithm, then it has a finite Littlestone dimension which is a contradiction.

The above result implies that for every $k$ and every learning algorithm $A$, there is *some* distribution $\mathcal{D}_A$ such that $A$ is not $k$-list replicable with respect to $\mathcal{D}_A$. However, a natural question is whether a lower bound holds for a fixed distribution, especially simple distributions such as the uniform distribution. We show that this is indeed the case.

**Theorem 6.7.** *In the PAC model under the uniform distribution, there is a $d + 1$-list replicable algorithm for the $d$-THRESHOLD. Moreover, for any $k < d+1$, there does not exist a $k$-list replicable algorithm for the concept class $d$-THRESHOLD under the uniform distribution. Thus its list complexity is exactly $d + 1$.*

# 7 Conclusions

In this work, we investigated the pressing issue of replicability in machine learning from an algorithmic point of view. We observed that perfect replicability is not achievable and hence considered two natural extensions that capture the degree of (non) replicability: list and certificate replicability. We designed replicable algorithms with a small list, certificate, and sample complexities for the $d$-COIN BIAS ESTIMATION PROBLEM and the class of problems that can be learned via statistical query algorithms that make non-adaptive statistical queries. We also established certain impossibility results in the PAC model of learning and for $d$-COIN BIAS ESTIMATION PROBLEM. There are several interesting research directions that emerge from our work. There is a gap in the sample complexities of the list and certificate replicable algorithms with comparable parameters. Is this gap inevitable? Currently, there is an exponential gap in the replicability parameters between hypothesis classes that can be learned via non-adaptive and adaptive statistical queries. Is this gap necessary? A generic question is to explore the trade-offs between the sample complexities, list complexity, certificate complexities, adaptivity, and nonadaptivity.

# 8 Acknowledgements

We thank the NeurIPS 2023 reviewers for their comments that improved the presentation of the paper. We especially acknowledge reviewer U7Lz for very detailed and insightful comments. We thank an anonymous researcher for pointing out Observation 6.6. We thank Jamie Radcliffe for informative discussions on topics related to this work. We thank authors of (CMY23) for informing us about their work and helpful discussions. Part of this work is supported by NSF grants 2130536 and 2130608. Part of the work was done during Pavan and Vinodchandran's visit to Simons Institute for the Theory of Computing. Part of Peter's work was done while at Ben-Gurion University of Negev.

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

# A   Appendix

## A.1   Missing Proofs from Section 4

To establish Lemmas 4.1 and 4.3, we build on the work of (VWDP$^+$22) and (DLPES02). We first introduce the necessary notion and definitions.

We recall the following notation from the main body of the paper. We use $\mathrm{diam}_\infty$ to indicate the diameter of a set relative to the $\ell_\infty$ norm and $\overline{B}_\varepsilon^\infty(\vec{p})$ to represent the closed ball of radius $\varepsilon$ centered at $\vec{p}$ relative to the $\ell_\infty$ norm. That is, in $\mathbb{R}^d$ we have $\overline{B}_\varepsilon^\infty(\vec{p}) = \prod_{i=1}^d [p_i - \varepsilon, p_i + \varepsilon]$.

Lemma 4.1 is based on the construction of certain geometric partitions of $\mathbb{R}^d$ called *secluded partitions*. Such partitions naturally induce deterministic rounding schemes which we use in the proof.

Let $\mathcal{P}$ be a partition of $\mathbb{R}^d$. For a point $\vec{p} \in \mathbb{R}^d$, let $N_\varepsilon(\vec{p})$ denote the set of members of the partitions that have a non-empty intersection with the $\varepsilon$-ball around $\vec{p}$. That is,

$$N_\varepsilon(\vec{p}) = \{X \in \mathbb{P} \mid \overline{B}_\varepsilon^\infty(\vec{p})) \cap X \neq \emptyset\}$$

**Definition A.1** (Secluded Partition). Let $\mathcal{P}$ be a partition of $\mathbb{R}^d$. We say that $\mathcal{P}$ is $(k, \varepsilon)$-*secluded*, if for every point $\vec{p} \in \mathbb{R}^d$, $|N_{\varepsilon(\vec{p})}| \leq k$.

The following theorem from (VWDP$^+$22) gives an explicit construction of a secluded partition with desired parameters where each member of the partition is a hypercube. For such partitions, we use the following notation. For every $\vec{p} \in \mathbb{R}^d$, if $\vec{p} \in \mathbb{P}$, then the *representative of $\vec{p}$*, $rep(\vec{p})$, is the center of the hypercube $X$.

**Theorem A.2.** *For each $d \in \mathbb{N}$, there exists a $(d+1, \frac{1}{2d})$-secluded partition, where each member of the partition is a unit hypercube. Moreover, the partition is efficiently computable: Given an arbitrary point $\vec{x} \in \mathbb{R}^d$, its representative can be computed in time polynomial in $d$.*

### A.1.1   Proof of Lemma 4.1

**Lemma A.3** (Lemma 4.1). *Let $d \in \mathbb{N}$ and $\varepsilon \in (0, \infty)$. Let $\varepsilon_0 = \frac{\varepsilon}{2d}$. There is an efficiently computable function $f_\varepsilon : \mathbb{R}^d \to \mathbb{R}^d$ with the following two properties:*

    *1. For any $x \in \mathbb{R}^d$ and any $\hat{x} \in \overline{B}_{\varepsilon_0}^\infty(x)$ it holds that $f_\varepsilon(\hat{x}) \in \overline{B}_\varepsilon^\infty(x)$.*

    *2. For any $x \in \mathbb{R}^d$ the set $\left\{ f_\varepsilon(\hat{x}) \colon \hat{x} \in \overline{B}_{\varepsilon_0}^\infty(x) \right\}$ has cardinality at most $d+1$.*

As explained in the main body, intuitively, item (1) states that if $\hat{x}$ is an $\varepsilon_0$-approximation of $x$, then $f_\varepsilon(\hat{x})$ is an $\varepsilon$-approximation of $x$, and item (2) states that $f_\varepsilon$ maps every $\varepsilon_0$-approximation of $x$ to one of at most $d+1$ possible values.

*Proof.* A high-level idea behind the proof is explained in Figure 1. We scale the $(d+1, \frac{1}{2d})$-secluded unit hypercube partition by $\varepsilon$ so that each partition member is a hypercube with side length $\varepsilon$. Now, for a point $x$, the ball $\overline{B}_{\varepsilon_0}^\infty(x)$ intersects at most $d+1$ hypercubes. Consider a point $\hat{x}_1 \in \overline{B}_{\varepsilon_0}^\infty(x)$, it is rounded to $c_1$ (center of the hypercube it resides in). Note that $c_1$ lies in the ball of radius $\varepsilon$ around $x$, this is because distance from $x$ to $\hat{x}_1$ is at most $\varepsilon_0$ and the distance from $\hat{x}_1$ to $c_1$ is at most $\varepsilon/2$. By triangle inequality $c_1$ belongs to $\overline{B}_\varepsilon^\infty(x)$. We now provide formal proof.

Let $\mathcal{P}$ be the $(d+1, \frac{1}{2d})$-secluded partition given by Theorem A.2. Thus $\mathcal{P}$ consists of unit cubes $[0, 1)^d$ with the property that for any point $\vec{p} \in \mathbb{R}^d$ the closed cube of side length $1/d$ centered at $\vec{p}$ (i.e. $\overline{B}_{\frac{1}{2d}}^\infty(\vec{p})$) intersects at most $d+1$ members/cubes of $\mathcal{P}$.

We first define a *rounding* function $f : \mathbb{R}^d \to \mathbb{R}^d$ as follows: for every $x \in \mathbb{R}^d$, $f(x) = rep(x)$.

Observe that the rounding function $f$ has the following two properties. (1) For every $x \in \mathbb{R}^d$, $\|f(x) - x\|_\infty \leq \frac{1}{2}$. This is because every point $x$ is mapped via $f$ to its representative, which is the center of the unit cube in which it lies. (2) For any point $\vec{p} \in \mathbb{R}^d$, the set $\left\{ f(x) \colon x \in \overline{B}_{\frac{1}{2d}}^\infty(\vec{p}) \right\}$ has

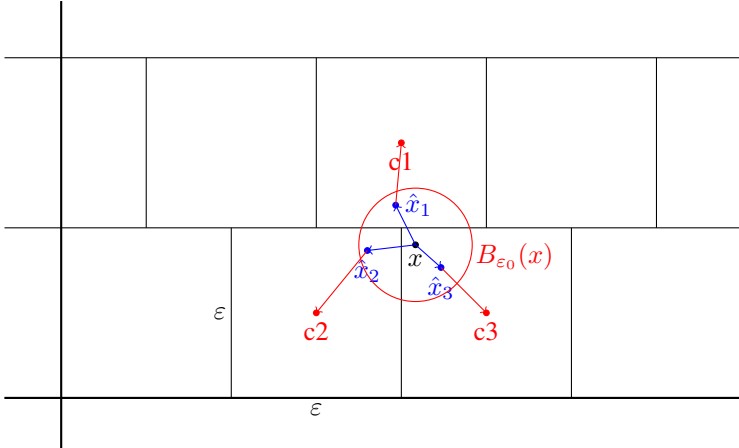

Figure 1: Illustration of proof of Lemma 4.1 for $d = 2$.

cardinality at most $d + 1$. This is because $\overline{B}^{\infty}_{\frac{1}{2d}}(\vec{p})$ intersects at most $d + 1$ hypercubes of $\mathcal{P}$ and for every hypercube $X$, all the points in $X$ are mapped to its center by $f$.

The function $f$ only gives an $\frac{1}{2}$-approximation guarantee. In order to get any $\varepsilon$-approximation guarantee, we scale $f$ appropriately. $f_{\varepsilon}$ is this *scaled* version of $f$.

Define the function $f_{\varepsilon} \colon \mathbb{R}^d \to \mathbb{R}^d$ as follows: for every $\hat{x} \in \mathbb{R}^d$, $f_{\varepsilon}(\hat{x}) = \varepsilon \cdot f(\frac{1}{\varepsilon}\hat{x})$. The efficient computability of $f_{\varepsilon}$ comes from the efficient computability of $f$.

We first establish that $f_{\varepsilon}$ has property (1) stated in the Lemma. Let $x \in \mathbb{R}^d$ and $\hat{x} \in \overline{B}^{\infty}_{\varepsilon_0}(x)$. Then we have the following (justifications will follow):

$$\begin{aligned}
\left\| \tfrac{1}{\varepsilon} \cdot f_{\varepsilon}(\hat{x}) - \tfrac{1}{\varepsilon}x \right\|_{\infty} &= \left\| f(\tfrac{1}{\varepsilon}\hat{x}) - \tfrac{1}{\varepsilon}x \right\|_{\infty} \\
&\leq \left\| f(\tfrac{1}{\varepsilon}\hat{x}) - \tfrac{1}{\varepsilon}\hat{x} \right\|_{\infty} + \left\| \tfrac{1}{\varepsilon}\hat{x} - \tfrac{1}{\varepsilon}x \right\|_{\infty} \\
&\leq \left\| f(\tfrac{1}{\varepsilon}\hat{x}) - \tfrac{1}{\varepsilon}\hat{x} \right\|_{\infty} + \tfrac{1}{\varepsilon}\|\hat{x} - x\|_{\infty} \\
&\leq \tfrac{1}{2} + \tfrac{1}{\varepsilon}\varepsilon_0 \\
&= \tfrac{1}{2} + \tfrac{1}{2d} \leq 1
\end{aligned}$$

The first line is by the definition of $f_{\varepsilon}$, the second is the triangle inequality, the third is scaling of norms, the fourth uses the property of $f$ that points are not mapped a distance more than $\frac{1}{2}$ along with the hypothesis that $\hat{x} \in \overline{B}^{\infty}_{\varepsilon_0}(x)$, the fifth uses the definition of $\varepsilon_0$, and the sixth uses the fact that $d \geq 1$.

Scaling both sides by $\varepsilon$ and using the scaling of norms, the above gives us $\|f_{\varepsilon}(\hat{x}) - x\|_{\infty} \leq \varepsilon$ which proves property (1) of the lemma.

To see that $f_{\varepsilon}$ has property (2), let $x \in \mathbb{R}^d$. We have the following set equalities:

$$\begin{aligned}
\left\{ f_{\varepsilon}(\hat{x}) \colon \hat{x} \in \overline{B}^{\infty}_{\varepsilon_0}(x) \right\} &= \left\{ \varepsilon \cdot f(\tfrac{1}{\varepsilon}\hat{x}) \colon \hat{x} \in \overline{B}^{\infty}_{\varepsilon_0}(x) \right\} \\
&= \left\{ \varepsilon \cdot f(a) \colon a \in \overline{B}^{\infty}_{\frac{1}{\varepsilon}\varepsilon_0}(\tfrac{1}{\varepsilon}x) \right\} \\
&= \left\{ \varepsilon \cdot f(a) \colon a \in \overline{B}^{\infty}_{\frac{1}{2d}}(\tfrac{1}{\varepsilon}x) \right\}
\end{aligned}$$

The first line is from the definition of $f_{\varepsilon}$, the second line is from re-scaling, and the third is from the definition of $\varepsilon_0$.

Because $f$ takes on at most $d + 1$ distinct values on $\overline{B}^{\infty}_{\frac{1}{2d}}(\tfrac{1}{\varepsilon}x)$, the set above has cardinality at most $d + 1$ which proves property (2) of the lemma. $\qquad\square$

### A.1.2 Proof of Lemma 4.2

**Lemma A.4** (Lemma 4.2). *Let $d \in \mathbb{N}$, $\varepsilon_0 \in (0, \infty)$ and $0 < \delta < 1$. There is an efficiently computable deterministic function $f : \{0,1\}^\ell \times \mathbb{R}^d \to \mathbb{R}^d$ with the following property. For any $x \in \mathbb{R}^d$,*

$$\Pr_{r \in \{0,1\}^\ell} \left[ \exists x^* \in \overline{B}_\varepsilon^\infty(x) \ \forall \hat{x} \in \overline{B}_{\varepsilon_0}^\infty(x) : f(r, \hat{x}) = x^* \right] \geq 1 - \delta$$

*where $\ell = \lceil \log \frac{d}{\delta} \rceil$ and $\varepsilon = (2^\ell + 1)\varepsilon_0 \leq \frac{2\varepsilon_0 d}{\delta}$.*

*Proof.* Partition each coordinate of $\mathbb{R}^d$ into $2\varepsilon_0$-width intervals. The algorithm computing the function $f$ does the following simple randomized rounding:

*The function $f$* : Choose a random integer $r \in \{1 \ldots 2^\ell\}$. Note that $r$ can be represented using $\ell$ bits. Consider the $i^{th}$ coordinate of $\hat{x}$ denoted by $\hat{x}[i]$. Round $\hat{x}[i]$ to the nearest $k * (2\varepsilon_0)$ such that $k \mod 2^\ell \equiv r$.

Now we will prove that $f$ satisfies the required properties.

First, we prove the approximation guarantee. Let $x'$ denote the point in $\mathbb{R}^d$ obtained after rounding each coordinate of $\hat{x}$. The $k$s satisfying $k \mod 2^\ell \equiv r$ are $2^\ell \cdot 2\varepsilon_0$ apart. Therefore, $x'[i]$ is rounded by at most $2^\ell \varepsilon_0$. That is, $|x'[i] - \hat{x}[i]| \leq 2^\ell \varepsilon_0 = \frac{\varepsilon_0 d}{\delta}$ for every $i$, $1 \leq i \leq d$. Since $\hat{x}$ is an $\varepsilon_0$-approximation (i.e. each coordinate $\hat{x}[i]$ is within $\varepsilon_0$ of the true value $x[i]$), then each coordinate of $x'$ is within $(2^\ell + 1)\varepsilon_0$ of $x[i]$. Therefore $x'$ is a $(2^\ell + 1)\varepsilon_0$-approximation of $x[i]$. Thus $x' \in \overline{B}_\varepsilon^\infty(x)$ for any choice of $r$.

Now we establish that for $\geq 1 - \delta$ fraction of $r \in \{1 \ldots 2^\ell\}$, there exists $x^*$ such every $\hat{x} \in \overline{B}_{\varepsilon_0}^\infty(x)$ is rounded $x^*$. We argue this with respect to each coordinate and apply the union bound. Fix an $x$ and a coordinate $i$. For $x[i]$, consider the $\varepsilon_0$ interval around it.

Consider $r$ from $\{1 \ldots 2^\ell\}$. When this $r$ is chosen, then we round $\hat{x}[i]$ to the closest $k * (2\varepsilon_0)$ such that $k \mod 2^\ell \equiv r$. Let $p_1^r, p_2^r, \ldots p_j^r \ldots$ be the set of such points: more precisely $p_j = (j2^l + r) * 2\varepsilon_0$. Note that $\hat{x}[i]$ is rounded to an $p_j$ to some $j$. Let $m_j^r$ denote the midpoint between $p_j^r$ and $p_{j+1}^r$. I.e, $m^r = (p_j^r + p_{j+1}^r)/2$ We call $r$ 'bad' for $x[i]$ if $x[i]$ is close to some $m_j^r$. That is, $r$ is 'bad' if $|x[i] - m_j^r| < \varepsilon_0$. Note that for a bad $r$ there exist $\hat{x_1}$ and $\hat{x_2}$ in $\overline{B}_{\varepsilon_0}^\infty(x)$ so that their $i^{th}$ coordinates are rounded to $p_j^r$ and $p_{j+1}^r$ respectively. The crucial point is that if $r$ is 'not bad' for $x[i]$, then for every $x' \in \overline{B}_{\varepsilon_0}^\infty(x)$, there exists a canonical $p^*$ such that $x'[i]$ is rounded to $p^*$. We call $r$ bad for $x$, if $r$ is bad for $x$, if there exists at least one $i$, $1 \leq i \leq d$ such that $r$ is bad for $x[i]$. With this, it follows that if $r$ is not bad for $x$, then there exists a canonical $x^*$ such that every $x' \in \overline{B}_{\varepsilon_0}^\infty(x)$ is rounded to $x^*$.

With this, the goal is to bound the probability that a randomly chosen $r$ is bad for $x$. For this, we first bound the probability that $r$ is bad for $x[i]$. We will argue that there exists at most one bad $r$ for $x[i]$. Suppose that there exist two numbers $r_1$ and $r_2$ that are both bad for $x[i]$. This means that $|x[i] - m_{j_1}^{r_1}| < \varepsilon_0$ and $|x[i] - m_{j_2}^{r_2}| < \varepsilon_0$ for some $j_1$ and $j_2$. Thus by triangle inequality $|m_{j_1}^{r_1} - m_{j_2}^{r_2}| < 2\varepsilon_0$. However, note that $|p_{j_1}^{r_1} - p_{j_2}^{r_2}|$ is $|(j_1 - j_2)2^\ell + (r_1 - r_2)|2\varepsilon_0$. Since $r_1 \neq r_2$, this value is at least $2\varepsilon_0$. This implies that the absolute value of difference between $m_{j_1}^{r_1}$ and $m_{j_2}^{r_2}$ is at least $2\varepsilon$ leading to a contradiction.

Thus the probability that $r$ is bad for $x[i]$ is at most $\frac{1}{2^\ell}$ and by the union bound the probability that $r$ is bad for $x$ is at most $\frac{d}{2^\ell} \leq \delta$. This completes the proof. $\square$

### A.1.3 Proof of Lemma 4.3

The proof, which is based on Sperner/KKM Lemma, is present in (VWDP$^+$22). Since our setting is slightly different, for completeness we give a proof.

We first introduce the necessary definitions and notation.

**Definition A.5** (Sperner/KKM Coloring). Let $d \in \mathbb{N}$ and $V = \{0,1\}^d$ denote a set of colors (which is exactly the set of vertices of $[0,1]^d$ so that colors and vertices are identified). Let $\chi : [0,1]^d \to V$

be a coloring function such that for any face $F$ of $[0,1]^d$ (of any dimension), for any $x \in F$, it holds that $\chi(x) \in V(F)$ where $V(F)$ is the vertex set of $F$ (informally, the color of $x$ is one of the vertices in the face $F$). Such a function $\chi$ will be called a Sperner/KKM coloring.

**Theorem A.6** (Cubical Sperner/KKM lemma (DLPES02)). *Let $d \in \mathbb{N}$ and $V = \{0,1\}^d$ and $\chi : [0,1]^d \to V$ be a Sperner/KKM coloring. Then there exists a subset $J \subset V$ with $|J| = d + 1$ and a point $\vec{y} \in [0,1]^d$ such that for all $j \in J$, $\vec{y} \in \overline{\chi^{-1}(j)}$ (informally, there is a point $\vec{y}$ that is touching at least $d + 1$ different colors).*

We will need to relate partitions to Sperner/KKM coloring so that we can use the Sperner/KKM Lemma.

For any co-ordinate $i$, let $\pi$ denote the standard projection map: $\pi_i : [0,1]^d \to [0,1]$ defined by $\pi_i(x) \overset{\text{def}}{=} x_i$ which maps $d$-dimensional points to the $i^{th}$ coordinate value. We extend this to sets: $\pi_i(X) = \{\pi_i(x) : x \in X\}$.

**Definition A.7** (Non-Spanning partition). Let $d \in \mathbb{N}$ and $\mathcal{P}$ be a partition of $[0,1]^d$. We say that $\mathcal{P}$ is a non-spanning partition if it holds for all $X \in \mathcal{P}$ and for all $i \in [d]$ that either $\pi_i(X) \not\ni 0$ or $\pi_i(X) \not\ni 1$ (or both).

Next, we state a lemma that asserts that for any non-spanning partition, there is a Sperner/KKM coloring that respects the partition: that is every member gets the same color.

**Lemma A.8** (Coloring Admission). *Let $d \in \mathbb{N}$, and $V = \{0,1\}^d$, and $\mathcal{P}$ a non-spanning partition of $[0,1]^d$. Then there exists a Sperner/KKM coloring $\chi : [0,1]^d \to V$ such that for every $X \in \mathcal{P}$, for every $x, y \in X$, $\chi(x) = \chi(y)$.*

Now we are ready to prove the Lemma 4.3.

**Lemma A.9.** *(Lemma 4.3) Let $\mathcal{P}$ be a partition of $[0,1]^d$ such that for each member $X \in \mathcal{P}$, it holds that $\mathrm{diam}_\infty(X) < 1$. Then there exists $\vec{p} \in [0,1]^d$ such that for all $\delta > 0$ we have that $\overline{B}_\delta^\infty(\vec{p})$ intersects at least $d + 1$ members of $\mathcal{P}$.*

*Proof.* Consider an arbitrary $X \in \mathcal{P}$. For each coordinate, $i \in [d]$, the set $\{x_i : x \in X\}$ does not contain both 0 and 1 (if it did, this would demonstrate two points in $X$ that are $\ell_\infty$ distance at least 1 apart and contradict that $\mathrm{diam}_\infty(X) < 1$). Thus, $\mathcal{P}$ is by definition a non-spanning partition of $[0,1]^d$. Since $\mathcal{P}$ is non-spanning, by Lemma A.8, there is a Sperner/KKM coloring where each point of $[0,1]^d$ can be assigned one of $2^d$-many colors and for any member $X \in \mathcal{P}$, all points in $X$ are assigned the same color. By Lemma A.6, there is a point $\vec{p} \in [0,1]^d$ such that $\vec{p}$ belongs to the closure of at least $d + 1$ colors. Since every point of a partition has the same color, each of these $d + 1$ colors corresponds to at least $d + 1$ different partitions. From this, it follows that or any $\delta > 0$, $\overline{B}_\delta^\infty(\vec{p})$ intersects at least $d + 1$ different members of $\mathcal{P}$. $\qquad\square$

## A.2 Missing Proofs From Section 6

### A.2.1 Proofs of Theorem 6.2, Theorem 6.3

**Theorem A.10** (Theorem 6.2). *Let $\mathcal{H}$ be a concept class that is learnable with $d$ non-adaptive statistical queries, then $\mathcal{H}$ is $(d + 1)$-list replicably learnable. Furthermore, the sample complexity $n = n(\nu, \delta)$ of the $(d + 1)$-list replicable algorithm is $O(\frac{d^2}{\nu^2} \cdot \log \frac{d}{\delta})$, where $\nu$ is the approximation error parameter of each statistical query oracle.*

*Proof.* The proof is very similar to the proof of Theorem 5.2. Our replicable algorithm $B$ works as follows. Let $\varepsilon$ and $\delta$ be input parameters and $\mathcal{D}$ be a distribution and $f \in \mathcal{H}$. Let $A$ be the statistical query learning algorithm for $\mathcal{H}$. Let $STAT(D_f, \nu)$ be the statistical query oracle for this algorithm. Let $\phi_1, \ldots, \phi_d$ be the statistical queries made by $A$.

Let $\vec{b} = \langle b[1], b[2], \ldots, b[d] \rangle$ where $b[i] = E_{\langle x,y \rangle \in \mathcal{D}_f}[\phi_i(\langle x, y \rangle)]$, $1 \le i \le d$. Set $\varepsilon_0 = \frac{\nu}{2d}$. The algorithm $B$ first estimates the values $b[i]$ up to an approximation error of $\varepsilon_0$ with success probably $1 - \delta/d$ for each query. Note that this can be done by a simple empirical estimation algorithm, that

uses a total of $n = O(\frac{d^2}{\nu^2} \cdot \log \frac{d}{\delta})$ samples. Let $\vec{v}$ be the estimated vector. It follows that $\vec{v} \in \overline{B}_{\varepsilon_0}^{\infty}(\vec{b})$ with probability at least $1 - \delta$. Note that different runs of the algorithm will output different $\vec{v}$.

Next, the algorithm $B$ evaluates the deterministic function $f_{\varepsilon}$ from Lemma 4.1 on input $\vec{v}$. Let $\vec{u}$ be the output vector. Finally, the algorithm $B$ simulates the statistical query algorithm $A$ with $\vec{u}[i]$ as the answer to the query $\phi_i$. By Lemma 4.1, $\vec{u} \in \overline{B}_{\nu}^{\infty}(\vec{b})$. Thus the error of the hypothesis output by the algorithm is at most $\varepsilon$. Since $A$ is a deterministic algorithm the number of possible outputs only depends on the number of outputs of the function $f_{\varepsilon}$, more precisely the number of possible outputs is the size of the set $\{f_{\varepsilon}(\vec{v}) : v \in \overline{B}_{\varepsilon_0}^{\infty}(\vec{b})\}$ which is almost $d + 1$, by Lemma 4.1. Thus the total number of possible outputs of the algorithm $B$ is at most $d + 1$ with probability at least $1 - \delta$. □

**Theorem A.11** (Theorem 6.3). *Let $\mathcal{H}$ be a concept class that is learnable with $d$ non-adaptive statistical queries, then $\mathcal{H}$ is $\lceil \log \frac{d}{\delta} \rceil$-certificate replicably learnable. Furthermore, the sample complexity $n = n(\nu, \delta)$ of this algorithm equals $O(\frac{d^2}{\nu^2 \delta^2} \cdot \log \frac{d}{\delta})$, where $\nu$ is the approximation error parameter of each statistical query oracle.*

*Proof.* The proof is very similar to the proof of Theorem 5.4. Our replicable algorithm $B$ works as follows, let $\varepsilon$ and $\delta$ be input parameters and $\mathcal{D}$ be a distribution and $f \in \mathcal{H}$. Let $A$ be the statistical query learning algorithm for $\mathcal{H}$ that outputs a hypothesis $h$ with approximation error $e_{\mathcal{D}_f}(h) = \varepsilon$. Let $STAT(D_f, \nu)$ be the statistical query oracle for this algorithm. Let $\phi_1, \ldots, \phi_d$ be the statistical queries made by $A$.

Let $\vec{b} = \langle b[1], b[2], \cdots, b[d] \rangle$, where $b[i] = E_{\langle x,y \rangle \in \mathcal{D}_f}[\phi_i(\langle x, y \rangle)]$. Set $\varepsilon_0 = \frac{\nu\delta}{2d}$. The algorithm $B$ first estimates the values $b[i], 1 \leq i \leq d$ up to an additive approximation error of $\varepsilon_0$ with success probably $1 - \delta/d$ for each query. Note that this can be done by a simple empirical estimation algorithm that uses a total of $n = O(\frac{d^2}{\nu^2 \delta^2} \cdot \log \frac{d}{\delta})$ samples. Let $\vec{v}$ be the estimated the vector. It follows that $\vec{v} \in \overline{B}_{\varepsilon_0}^{\infty}(\vec{b})$ with probability at least $1 - \delta$. Next, the algorithm $B$ evaluates the deterministic function $f$ described in Lemma 4.2 with inputs $r \in \{0, 1\}^{\ell}$ where $\ell = \lceil \log \frac{d}{\delta} \rceil$ and $\vec{v}$. By Lemma 4.2 for at least $1 - \delta$ fraction of the $r$'s , the function $f$ outputs a canonical $\vec{v^*} \in \overline{B}_{\nu}^{\infty}(\vec{b})$. Finally, the algorithm $B$ simulates the statistical query algorithm $A$ with $\vec{v^*}[i]$ as the answer to the query $\phi_i$. Since $A$ is a deterministic algorithm it follows that our algorithm $B$ is certificate replicable. Note that the certificate complexity is $\ell = \lceil \log \frac{d}{\delta} \rceil$. □

The following theorem states how to convert adaptive statistical query learning algorithms into certificate replicable PAC learning algorithms. This result also appears in the work of (GKM21; ILPS22), though they did not state the certificate complexity. We explicitly state the result here.

**Theorem A.12.** *((GKM21; ILPS22))[Theorem 6.4] Let $\mathcal{H}$ be a concept class that is learnable with $d$ adaptive statistical queries, then $\mathcal{H}$ is $\lceil d \log \frac{d}{\delta} \rceil$-certificate replicably learnable. Furthermore, the sample complexity of this algorithm equals $O(\frac{d^3}{\nu^2 \delta^2} \cdot \log \frac{d}{\delta})$, where $\nu$ is the approximation error parameter of each statistical query oracle.*

*Proof.* The proof uses similar arguments as before. The main difference is that we will evaluate each query with an approximation error of $\frac{\nu\delta}{d}$ with a probability error of $d/\delta$. This requires $O(\frac{d^2}{\nu^2 \delta^2} \cdot \log \frac{d}{\delta})$ per query. We use a fresh set of certificate randomness for each such evaluation. Note that the length of the certificate for each query is $\lceil \log d/\delta \rceil$. Thus the total certificate complexity is $\lceil d \log \frac{d}{\delta} \rceil$. □

### A.2.2  Proof of Theorem 6.7

We first recall the definition of the concept class $d$-THRESHOLD.

Fix some $d \in \mathbb{N}$. Let $X = [0, 1]^d$. For each value $\vec{t} \in [0, 1]^d$, let $h_{\vec{t}} : X \to \{0, 1\}$ be the concept defined as follows: $h_{\vec{t}}(\vec{x}) = 1$ if for every $i \in [d]$ it holds that $x_i \leq t_i$ and 0 otherwise. Let $\mathcal{H}$ be the hypothesis class consisting of all such threshold concepts: $\mathcal{H} = \{h_{\vec{t}} \mid \vec{t} \in [0, 1]^d\}$.

**Theorem A.13** (Theorem 6.7). *In the PAC model under the uniform distribution, there is a $d + 1$-list replicable algorithm for the $d$-THRESHOLD. Moreover, for any $k < d + 1$, there does not exist a $k$-list replicable algorithm for the concept class $d$-THRESHOLD under the uniform distribution. Thus its list complexity is exactly $d + 1$.*

It is easy to see that $d$-THRESHOLD is learnable under the uniform distribution by making $d$ non-adaptive statistical queries. Thus by Theorem 6.2, $d$-THRESHOLD under the uniform distribution admits a $(d+1)$-list replicable algorithm. So we will focus on proving the lower bound which is stated as a separate theorem below.

**Theorem A.14.** *For $k < d+1$, there does not exist a $k$-list replicable algorithm for the $d$-THRESHOLD in the PAC model under uniform distribution.*

The proof is similar to the proof of Theorem 5.5. The reason is that sampling $d$-many biased coins with bias vector $\vec{b}$ is similar to obtaining a point $\vec{x}$ uniformly at random from $[0,1]^d$ and evaluating the threshold function $h_{\vec{b}}$ on it—this corresponds to asking whether all of the coins were heads/1's. The two models differ though, because in the sample model for the $d$-COIN BIAS ESTIMATION PROBLEM, the algorithm sees for each coin whether it is heads or tails, but this information is not available in the PAC model for the $d$-THRESHOLD. Conversely, in the PAC model for the $d$-THRESHOLD, a random draw from $[0,1]^d$ is available to the algorithm, but in the sample model for the $d$-COIN BIAS ESTIMATION PROBLEM the algorithm does not get this information.

Furthermore, there is the following additional complexity in the impossibility result for the $d$-THRESHOLD. In the $d$-COIN BIAS ESTIMATION PROBLEM, we said by definition that a collection of $d$ coins parameterized by bias vector $\vec{a}$ was an $\varepsilon$-approximation to a collection of $d$ coins parameterized by bias vector $\vec{b}$ if and only if $\|\vec{b}-\vec{a}\|_\infty \leq \varepsilon$, and we used this norm in the proofs. However, the notion of $\varepsilon$-approximation in the PAC model is quite different than this. It is possible to have a hypotheses $h_{\vec{a}}$ and $h_{\vec{b}}$ in the $d$-THRESHOLD such that $\|\vec{b}-\vec{a}\|_\infty > \varepsilon$ but with respect to some distribution $\mathcal{D}_X$ on the domain $X$ we have $err_{\mathcal{D}_X}(h_{\vec{a}}, h_{\vec{b}}) \leq \varepsilon$. For example, if $\mathcal{D}_X$ is the uniform distribution on $X = [0,1]^d$ and $\vec{a} = \vec{0}$ and $\vec{b}$ is the first standard basis vector $\vec{b} = \langle 1, 0, \ldots, 0 \rangle$, and $\varepsilon = \frac{1}{2}$, then $\|\vec{b}-\vec{a}\|_\infty = 1 > \varepsilon$, but $err_{\mathcal{D}_X}(h_{\vec{a}}, h_{\vec{b}}) = 0 \leq \varepsilon$ because $h_{\vec{a}}(\vec{x}) \neq h_{\vec{b}}(\vec{x})$ if and only if all of the last $d-1$ coordinates of $\vec{x}$ are 0 and the first coordinate is $> 0$, but there is probability 0 of sampling such $\vec{x}$ from the uniform distribution on $X = [0,1]^d$.

For this reason, we can't just partition $[0,1]^d$ as we did with the proof of Theorem 5.5 and must do something more clever. It turns out that it is possible to find a subset $[\alpha, 1]^d$ on which hypotheses parameterized by vectors on opposite faces of this cube $[\alpha, 1]^d$ have high PAC error between them. A consequence by the triangle inequality of $err_{\mathcal{D}_X}$ is that two such hypotheses cannot both be approximated by a common third hypothesis. This is the following lemma states.

**Lemma A.15.** *Let $d \in \mathbb{N}$ and $\alpha = \frac{d-1}{d}$. Let $\vec{s}, \vec{t} \in [\alpha, 1]^d$ such that there exists a coordinate $i_0 \in [d]$ where $s_{i_0} = \alpha$ and $t_{i_0} = 1$ (i.e. $\vec{s}$ and $\vec{t}$ are on opposite faces of this cube). Let $\varepsilon \leq \frac{1}{8d}$. Then there is no point $\vec{r} \in X$ such that both $err_{\text{unif}}(h_{\vec{s}}, h_{\vec{r}}) \leq \varepsilon$ and $err_{\text{unif}}(h_{\vec{t}}, h_{\vec{r}}) \leq \varepsilon$ (i.e. there is no hypothesis which is an $\varepsilon$-approximation to both $h_{\vec{s}}$ and $h_{\vec{t}}$).*

*Proof.* Let $\vec{q} = \left\langle \begin{cases} s_i & i = i_0 \\ t_i & i \neq i_0 \end{cases} \right\rangle_{i=1}^d$ which will serve as a proxy to $\vec{s}$.

We need the following claim.

**Claim A.16.** *For each $\vec{x} \in X$, the following are equivalent:*

1. $h_{\vec{q}}(\vec{x}) \neq h_{\vec{t}}(\vec{x})$

2. $h_{\vec{q}}(\vec{x}) = 0$ *and* $h_{\vec{t}}(\vec{x}) = 1$

3. $x_{i_0} \in (q_{i_0}, t_{i_0}] = (\alpha, 1]$ *and for all* $i \in [d] \setminus \{i_0\}$, $x_i \in [0, t_i]$.

*Furthermore, the above equivalent conditions imply the following:*

4. $h_{\vec{s}}(\vec{x}) \neq h_{\vec{t}}(\vec{x})$.

*Proof of Claim A.16.*

$(2) \implies (1)$: This is trivial.

(1) $\Longrightarrow$ (2): Note that because $q_{i_0} = s_{i_0} = \alpha < 1 = t_{i_0}$, we have for all $i \in [d]$ that $q_i \leq t_i$. If $h_{\vec{t}}(\vec{x}) = 0$ then for some $i_1 \in [d]$ it must be that $x_{i_1} > t_{i_1}$, but since $t_{i_1} \geq q_{i_1}$ it would also be the case that $x_{i_1} > q_{i_1}$, so $h_{\vec{q}}(\vec{x}) = 0$ which gives the contradiction that $h_{\vec{q}}(\vec{x}) = h_{\vec{t}}(\vec{x})$. Thus $h_{\vec{t}}(\vec{x}) = 1$, and since $h_{\vec{q}}(\vec{x}) \neq h_{\vec{t}}(\vec{x})$ we have $h_{\vec{q}}(\vec{x}) = 0$.

(1) $\Longleftrightarrow$ (3): We partition $[0,1]^d$ into three sets and examine these three cases.

Case 1: $x_{i_0} \in (q_{i_0}, t_{i_0}] = (\alpha, 1]$ and for all $i \in [d] \setminus \{i_0\}$, $x_i \in [0, t_i]$. In this case, $q_{i_0} < x_{i_0}$ so $h_{\vec{q}}(\vec{x}) = 0$ and for all $i \in [d]$ $x_i \leq t_i$, so $h_{\vec{t}}(\vec{x}) = 1$, so $h_{\vec{q}}(\vec{x}) \neq h_{\vec{t}}(\vec{x})$.

Case 2: $x_{i_0} \notin (q_{i_0}, t_{i_0}] = (\alpha, 1]$ and for all $i \in [d] \setminus \{i_0\}$, $x_i \in [0, t_i]$. In this case, because $x_{i_0} \in [0,1]$ and $x_{i_0} \notin (\alpha, 1]$ we have $x_{i_0} \leq \alpha = q_{i_0} \leq t_{i_0}$ and also for all other $i \in [d] \setminus \{i_0\}$, $x_i \leq t_i = q_i$ (by definition of $\vec{q}$). Thus $h_{\vec{q}}(\vec{x}) = 1 = h_{\vec{t}}(\vec{x})$.

Case 3: For some $i_1 \in [d] \setminus \{i_0\}$, $x_{i_1} \notin [0, t_{i_1}]$. In this case, because $x_{i_1} \in [0,1]$, we have $x_{i_1} > t_{i_1} = q_{i_1}$. Thus $h_{\vec{q}}(\vec{x}) = 0 = h_{\vec{t}}(\vec{x})$.

Thus, it is the case that $h_{\vec{q}}(\vec{x}) \neq h_{\vec{t}}(\vec{x})$ if and only if $x_{i_0} \in (q_{i_0}, t_{i_0}] = (\alpha, 1]$ and for all $i \in [d] \setminus \{i_0\}$, $x_i \in [0, t_i]$.

(1, 2, 3) $\Longrightarrow$ (4): By (2), we have $x_{i_0} > q_{i_0}$, and since $q_{i_0} = s_{i_0}$ by definition of $\vec{q}$, it follows that $x_{i_0} > s_{i_0}$ which means $h_{\vec{s}}(\vec{x}) = 0$. By (3), $h_{\vec{t}}(\vec{x}) = 1$ which gives $h_{\vec{s}}(\vec{x}) \neq h_{\vec{t}}(\vec{x})$. $\qquad\square$

We also need the following Lemma.

**Lemma A.17.** *Let* $d \in \mathbb{N}$ *and* $\alpha = \frac{d-1}{d} = 1 - \frac{1}{d}$. *Then* $(1-\alpha) \cdot \alpha^{d-1} > \frac{1}{4d}$.

*Proof.* If $d = 1$, then $\alpha = 0$ so $(1-\alpha) \cdot \alpha^{d-1} = 1 \geq \frac{1}{4} = \frac{1}{4d}$ (see footnote[2]).

If $d \geq 2$, then we utilize the fact that $(1 - \frac{1}{d})^d \geq \frac{1}{4}$ in the following:

$$
\begin{aligned}
(1-\alpha) \cdot \alpha^{d-1} &= (\tfrac{1}{d})(1 - \tfrac{1}{d})^{d-1} \\
&= (\tfrac{1}{d}) \frac{(1 - \tfrac{1}{d})^d}{1 - \tfrac{1}{d}} \\
&= \frac{(1 - \tfrac{1}{d})^d}{d-1} \\
&\geq \frac{1}{4(d-1)} \\
&> \frac{1}{4d}.
\end{aligned}
$$

This completes the proof. As an aside, $\alpha = \frac{d-1}{d}$ is the value of $\alpha$ that maximizes the expression $(1-\alpha) \cdot \alpha^{d-1}$ which is why that value was chosen. $\qquad\square$

With the above Claim and Lemma in hand, we return to the proof of Lemma A.15. Our next step will be two prove the following two inequalities:

$$ 2\varepsilon < err_{\mathrm{unif}}(h_{\vec{q}}, h_{\vec{t}}) \leq err_{\mathrm{unif}}(h_{\vec{s}}, h_{\vec{t}}). $$

For the second of these inequalities, note that by the (1) $\Longrightarrow$ (4) part of claim above, since $h_{\vec{q}}(\vec{x}) \neq h_{\vec{t}}(\vec{x})$ implies $h_{\vec{s}}(\vec{x}) \neq h_{\vec{t}}(\vec{x})$ we have

$$
\begin{aligned}
err_{\mathrm{unif}}(h_{\vec{q}}, h_{\vec{t}}) &= \Pr_{\vec{x} \sim \mathrm{unif}(X)}[h_{\vec{q}}(\vec{x}) \neq h_{\vec{t}}(\vec{x})] \\
&\leq \Pr_{\vec{x} \sim \mathrm{unif}(X)}[h_{\vec{s}}(\vec{x}) \neq h_{\vec{t}}(\vec{x})] \\
&= err_{\mathrm{unif}}(h_{\vec{s}}, h_{\vec{t}}).
\end{aligned}
$$

---

[2]This uses the interpretation that $0^0 = 1$ which is the correct interpretation in the context in which we will use the lemma.

Now, for the first of the inequalities above, we will use the (1) $\iff$ (3) portion of the claim, we will use our hypothesis that $\vec{t} \in [\alpha, 1]^d$ (which implies for each $i \in [d]$ that $[0, t_i] \subseteq [0, \alpha]$), we will use the hypothesis that $\varepsilon \leq \frac{1}{8d}$, and we will use Theorem A.17. Utilizing these, we get the following:

$$
\begin{aligned}
&err_{\text{unif}}(h_{\vec{q}}, h_{\vec{t}}) \\
&= \Pr_{\vec{x} \sim \text{unif}(X)}[h_{\vec{q}}(\vec{x}) \neq h_{\vec{t}}(\vec{x})] \\
&= \Pr_{\vec{x} \sim \text{unif}(X)}[x_{i_0} \in (\alpha, 1] \ \wedge \ \forall i \in [d] \setminus \{i_0\}, \ x_i \in [0, t_i]] \\
&= \Pr_{x_{i_0} \sim \text{unif}([0,1])}[x_{i_0} \in (\alpha, 1]] \cdot \prod_{\substack{i=1 \\ i \neq i_0}}^{d} \Pr_{x \sim \text{unif}([0,1])}[x \in [0, t_i]] \\
&\geq \Pr_{x_{i_0} \sim \text{unif}([0,1])}[x_{i_0} \in (\alpha, 1]] \cdot \prod_{\substack{i=1 \\ i \neq i_0}}^{d} \Pr_{x \sim \text{unif}([0,1])}[x \in [0, \alpha]] \\
&= (1 - \alpha) \cdot \alpha^{d-1} \\
&> \frac{1}{4d} \\
&\geq 2\varepsilon.
\end{aligned}
$$

Thus, we get the desired two inequalities:

$$
2\varepsilon < err_{\text{unif}}(h_{\vec{q}}, h_{\vec{t}}) \leq err_{\text{unif}}(h_{\vec{s}}, h_{\vec{t}}).
$$

This nearly completes the proof. If there existed some point $\vec{r} \in X$ such that both $err_{\text{unif}}(h_{\vec{s}}, h_{\vec{r}}) \leq \varepsilon$ and $err_{\text{unif}}(h_{\vec{t}}, h_{\vec{r}}) \leq \varepsilon$, then it would follow from the triangle inequality of $err_{\text{unif}}$ that

$$
err_{\text{unif}}(h_{\vec{s}}, h_{\vec{t}}) \leq err_{\text{unif}}(h_{\vec{s}}, h_{\vec{r}}) + err_{\text{unif}}(h_{\vec{t}}, h_{\vec{r}}) \leq 2\varepsilon
$$

but this would contradict the above inequalities, so no such $\vec{r}$ exists. $\qquad \square$

Equipped with the Lemma A.15, we are now ready to prove Theorem A.14.

*Proof of Theorem A.14.* Fix any $d \in \mathbb{N}$, and choose $\varepsilon$ and $\delta$ as $\varepsilon \leq \frac{1}{4d}$ and $\delta \leq \frac{1}{d+2}$. We will use the constant $\alpha = \frac{d-1}{d}$ and consider the cube $[\alpha, 1]^d$.

Suppose for contradiction such an algorithm $A$ does exists for some $k < d + 1$. This means that for each possible threshold $\vec{t} \in [0, 1]^d$, there exists some set $L_{\vec{t}} \subseteq \mathcal{H}$ of hypotheses with three properties: (1) each element of $L_{\vec{t}}$ is an $\varepsilon$-approximation to $h_{\vec{t}}$, (2) $|L_{\vec{t}}| \leq k$, and (3) with probability at least $1 - \delta$, $A$ returns an element of $L_{\vec{t}}$.

By the trivial averaging argument, this means that there exists at least one element in $L_{\vec{t}}$ which is returned by $A$ with probability at least $\frac{1}{k} \cdot (1 - \delta) \geq \frac{1}{k} \cdot (1 - \frac{1}{d+2}) = \frac{1}{k} \cdot \frac{d+1}{d+2} \geq \frac{1}{k} \cdot \frac{k+1}{k+2}$. Let $f : [\alpha, 1]^d \rightarrow [0, 1]^d$ be a function which maps each threshold $\vec{t} \in [\alpha, 1]^d$ to such an element (the maximum probability element with ties broken arbitrarily) of $L_{\vec{t}}$. This is slightly different from the proof of Theorem 5.5 because we are defining the function $f$ on only a very specific subset of the possible thresholds. The reason for this was alluded to in the discussion following the statement of Theorem A.14.

The function $f$ induces a partition $\mathcal{P}$ of $[\alpha, 1]^d$ where the members of $\mathcal{P}$ are the fibers of $f$ (i.e. $\mathcal{P} = \{f^{-1}(\vec{y}) : \vec{y} \in \text{range}(f)\}$). For any member $W \in \mathcal{P}$ and any coordinate $i \in [d]$, it cannot be that the set $\{w_i : \vec{w} \in W\}$ contains both values $\alpha$ and $1$—if it did, then there would be two points $\vec{s}, \vec{t} \in W$ such that $s_i = \alpha$ and $t_i = 1$, but because they both belong to $W$, there is some $\vec{y} \in [0, 1]^d$ such that $f(\vec{s}) = \vec{y} = f(\vec{t})$, but by definition of the partition, $h_{\vec{y}}$ would have to be an $\varepsilon$-approximation (in the PAC model) of both $h_{\vec{s}}$ and $h_{\vec{t}}$, but by Lemma A.15 this is not possible.

Thus, the partition $\mathcal{P}$ is a *non-spanning* partition of $[\alpha, 1]^d$ as in the proof of Lemma 4.3, so there is some point $\vec{p} \in [\alpha, 1]^d$ such that for every radius $r > 0$, it holds that $\overline{B}_r^{\infty}(\vec{p})$ intersects at least $d + 1$

members of $\mathcal{P}$. In fact, there is some radius $r$ such that $\|\vec{t} - \vec{s}\|_\infty \leq r$, then $d_{\mathrm{TV}}(\mathcal{D}_{A,\vec{s},n}, \mathcal{D}_{A,\vec{t},n}) \leq \eta$, for $\eta$ that lies between 0 and $\frac{1}{k} \cdot \frac{k+1}{k+2} - \frac{1}{k+1}$.

Now we get the same type of contradiction as in the proof of Theorem 5.5: for the special point $\vec{p}$ we have that $\mathcal{D}_{A,\vec{p},n}$ is a distribution that has $d + 1 \geq k + 1$ disjoint events that each have probability greater than $\frac{1}{k+1}$. Thus, no $k$-list replicable algorithm exists. $\qquad\square$

# B  Expanded Prior and Related Work

We give a more detailed discussion on prior and related work. This section is an elaboration of the Section 2 from the main body of the paper.

Formalizing reproducibility and replicability has gained considerable momentum in recent years. While the terms reproducibility and replicability are very close and often used interchangeably, there has been an effort to distinguish between them and accordingly, our notions fall in the replicability definition (PVLS$^+$21).

In the context of randomized algorithms, various notions of reproducibility/replicability have been investigated. The work of Gat and Goldwasser (GG11) formalized and defined the notion of *pseudodeterministic algorithms*. A randomized algorithm $A$ is *pseudodeterministic* if, for any input $x$, there is a canonical value $v_x$ such that $\Pr[A(x) = v_x] \geq 2/3$. Gat and Goldwasser designed polynomial-time pseudodeterministic algorithms for algebraic computational problems, such as finding quadratic non-residues and finding non-roots of multivariate polynomials (GG11). Later works studied the notion of pseudodeterminism in other algorithmic settings, such as parallel computation, streaming and sub-linear algorithms, interactive proofs, and its connections to complexity theory (GG; GGH18; OS17; OS18; AV20; GGMW20; LOS21; DPVWV22).

In the algorithmic setting, mainly two generalizations of pseudodeterminism have been investigated: *multi-pseudodeterministic algorithms* (Gol19) and *influential bit algorithms* (GL19). A randomized algorithm $A$ is $k$-pseudodeterministic if, for every input $x$, there is a set $S_x$ of size at most $k$ such that the output of $A(x)$ belongs to the set $S_x$ with high probability. When $k = 1$, we get pseudodeterminism. A randomized algorithm $A$ is $\ell$-influential-bit algorithm if, for every input $x$, for most of the strings $r$ of length $\ell$, there exists a canonical value $v_{x,r}$ such that the algorithm $A$ on inputs $x$ and $r$ outputs $v_{x,r}$ with high probability. The string $r$ is called the *influential bit* string. Again, when $\ell = 0$, we get back pseudodeterminism. The main focus of these works has been to investigate reproducibility in randomized search algorithms.

Very recently, pseudodeterminism and its generalizations have been explored in the context of learning algorithms to formalize the notion of replicability. The seminal work of (BLM20) defined the notion of *global stability*. They define a learning algorithm $A$ to be $(n, \eta)$-globally stable with respect to a distribution $D$ if there is a hypothesis $h$ such that $\Pr_{S \sim D^n}(A(S) = h) \geq \eta$, here $\eta$ is called the *stability parameter*. Note that the notion of global stability is equivalent to Gat and Goldwasser's notion of pseudodeterminism when $\eta = 2/3$. Since Gat and Goldwasser's motivation is to study pseudodeterminism in the context of randomized algorithms, the success probability is taken as $2/3$. In the context of learning, studying the stability parameter $\eta$ turned out to be useful. The work of Bun, Livny and Moran (BLM20) showed that any concept class with Littlestone dimension $d$ has an $(m, \eta)$-globally stable learning algorithm with $m = \tilde{O}(2^{2^d}/\alpha)$ and $\eta = \tilde{O}(2^{-2^d})$, where the error of $h$ (with respect to the unknown hypothesis) is $\leq \alpha$. Then they established that a globally stable learner implies a differentially private learner. This, together with an earlier work of Alon, Livny, Malliaris, and Moran (ALMM19), establishes an equivalence between online learnability and differentially private PAC learnability.

The work of Ghazi, Kumar, and Manurangsi (GKM21) extended the notion of global stability to pseudo-global stability and list-global stability. The notion of pseudo-global stability is very similar to the earlier-mentioned notion of influential bit algorithms of Grossman and Liu (GL19) when translated to the context of learning. The work of (GKM21) used these concepts to design user-level differentially private algorithms.

The recent work reported in (ILPS22) introduced the notion of $\rho$-*replicability*. A learning algorithm $A$ is $\rho$-replicable if $\Pr_{S_1,S_2,r}[A(S_1, r) = A(S_2, r)] \geq 1 - \rho$, where $S_1$ and $S_2$ are samples drawn from a distribution $\mathcal{D}$ and $r$ is the internal randomness of the learning algorithm $A$. They designed replicable

algorithms for many learning tasks, including statistical queries, approximate heavy hitters, median, and learning half-spaces. It is known that the notions of pseudo-global stability and $\rho$-replicability are the same up to polynomial factors in the parameters (ILPS22; GKM21).

In this work, we study the notions of list and certificate complexities as a measure the *degree of (non) replicability*. Our goal is to design learning algorithms with optimal list and certificate complexities while minimizing the sample complexity. The earlier works (BLM20; GKM21; ILPS22) did not focus on minimizing these quantities. The works of (BLM20; GKM21) used replicable algorithms as an intermediate step to design differentially private algorithms. The work of (ILPS22) did not consider reducing the certificate complexity in their algorithms and also did not study list-replicability. Earlier works (GKM21; ILPS22) studied how to convert statistical query learning algorithms into certificate replicable learning algorithms, however, their focus was not on the certificate complexity. Here, we study the relationship among (nonadaptive and adaptive) statistical query learning algorithms, list replicable algorithms, and certificate replicable algorithms with a focus on list, certificate and sample complexities.

A very recent and independent work of (CMY23) investigated relations between list replicability and the stability parameter $\nu$, in the context of distribution-free PAC learning. They showed that for every concept class $\mathcal{H}$, its list complexity is exactly the inverse of the stability parameter. They also showed that the list complexity of a hypothesis class is at least its VC dimension. For establishing this they exhibited, for any $d$, a concept class whose list complexity is exactly $d$. There are some similarities between their work and the present work. We establish similar upper and lower bounds on the list complexity but for different learning tasks: $d$-THRESHOLD and $d$-COIN BIAS ESTIMATION PROBLEM. For $d$-THRESHOLD, our results are for PAC learning under *uniform* distribution and do not follow from their distribution-independent results. Thus our results, though similar in spirit, are incomparable to theirs. Moreover, their work did not focus on efficiency in sample complexity and also did not study certificate complexity which is a focus of our paper. We do not study the stability parameter.

The study of notions of reproducibility/replicability in various computational fields is an emerging topic. The article (PVLS$^+$21) discusses the differences between replicability and reproducibility. In (EKK$^+$23), the authors consider replicability in the context of stochastic bandits. Their notion is similar to the notion studied in (ILPS22). In (AJJ$^+$22), the authors investigate *reproducibility* in the context of optimization with *inexact oracles* (initialization/gradient oracles). The setup and focus of these works are different from ours.

Finally, we would like to point out that there notions of list PAC learning and list-decodable learning in the learning theory literature (see (CP23) and (RY20) for recent progress on these notions). However, these notions are different from the list replicable learning that we consider in this paper. List PAC learning and list-decodable learning are generalized models of PAC learning. For example, any learning task that is PAC learnable is trivially list PAC learnable with a list size of 1. However, list replicable learning is an additional requirement that needs to be satisfied by a learner. Thus the notion of list and list-decodable PAC learning are different from the notions of list/certificate replicability.

