## A  Supplementary Material: Proofs

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

. Similarly, the list-global stability is similar to Goldreich's notion of multi-pseudodeterminism (Gol19). These notions coincide with our definitions of list replicability and certificate replicability respectively. The work of (GKM21) used these concepts to design user-level differentially private algorithms.

The recent work reported in (ILPS22) introduced the notion of $\rho$-*replicability*. A learning algorithm $A$ is $\rho$-replicable if $\Pr_{S_1, S_2, r}[A(S_1, r) = A(S_2, r)] \geq 1 - \rho$, where $S_1$ and $S_2$ are samples drawn from a distribution $\mathcal{D}$ and $r$ is the internal randomness of the learning algorithm $A$. They designed replicable algorithms for many learning tasks, including statistical queries, approximate heavy hitters, median, and learning half-spaces. It is known that the notions of pseudo-global stability and $\rho$-replicability are the same up to polynomial factors in the parameters (ILPS22; GKM21).

In this work, we study the notions of list and certificate complexities as a measure the *degree of (non) replicability*. Our goal is to design learning algorithms with optimal list and certificate complexities while minimizing the sample complexity. The earlier works (BLM20; GKM21; ILPS22) did not focus on minimizing these quantities. The works of (BLM20; GKM21) used replicable algorithms as an intermediate step to design differentially private algorithms. The work of (ILPS22) did not consider reducing the certificate complexity in their algorithms and also did not study list-replicability. Earlier works (GKM21; ILPS22) studied how to convert statistical query learning algorithms into certificate replicable learning algorithms, however, their focus was not on the certificate complexity. Here, we study the relationship among (nonadaptive and adaptive) statistical query learning algorithms, list replicable algorithms, and certificate replicable algorithms with a focus on list, certificate and sample complexities.

A very recent and independent work of (CMY23) investigated relations between list replicability and the stability parameter $\nu$, in the context of distribution-free PAC learning. They showed that for every concept class $\mathcal{H}$, its list complexity is exactly the inverse of the stability parameter. They also showed that the list complexity of a hypothesis class is at least its VC dimension. For establishing this they exhibited, for any $d$, a concept class whose list complexity is exactly $d$. There are some similarities between their work and the present work. We establish similar upper and lower bounds on the list complexity but for different learning tasks: $d$-THRESHOLD and $d$-COIN BIAS ESTIMATION PROBLEM. For $d$-THRESHOLD, our results are for PAC learning under *uniform* distribution and do not follow from their distribution-independent results. Thus our results, though similar in spirit, are incomparable to theirs. Moreover, their work did not focus on efficiency in sample complexity and also did not study certificate complexity which is a focus of our paper. We do not study the stability parameter.

The study of notions of reproducibility/replicability in various computational fields is an emerging topic. The article (PVLS$^+$21) discusses the differences between replicability and reproducibility. In

787 ([EKK+23), the authors consider replicability in the context of stochastic bandits. Their notion is
788 similar to the notion studied in (ILPS22). In (AJJ+22), the authors investigate *reproducibility* in the
789 context of optimization with *inexact oracles* (initialization/gradient oracles). The setup and focus of
790 these works are different from ours.