# OpenReview forum: "List and Certificate Complexities in Replicable Learning"
_NeurIPS.cc/2023/Conference — NeurIPS 2023 spotlight_

### Official Review · Reviewer_U7Lz · 2023-07-02

**Soundness:** 4 excellent
**Presentation:** 3 good
**Contribution:** 3 good
**Rating:** 7
**Confidence:** 5

**Summary:**

In this work the authors study the problem of replicability in machine learning from a mathematical point of view. They first show that even in the simple problem of estimating the bias of a coin, a strong notion of replicability which asks for an algorithm to output the same answer on independent i.i.d. samples from the true coin as input (and independent internal randomness) is not possible, unless the accuracy guarantees are trivial. Motivated by that, they propose two definitions that aim to capture the degree of non-replicability of the underlying statistical task.
 * The first one is called $L$ list-replicability and, roughly speaking, asks that the algorithm outputs at most $L$ different answers under independent executions, with high probability. Moreover, the answers that belong to this list need to be $\epsilon$-accurate for the specific application.
  * The second definition they propose is called $\ell$ certificate replicability. Under this definition, the algorithm has access to internal random strings of length $\ell$, and it is required that, for most of the random strings there is a ``canonical'' output $\mathcal{o}_r$ that is $\varepsilon$-accurate, i.e., with high probability over the draw of the dataset, when the random string of the algorithm is $r$ its output will be $\mathcal{o}_r$.

The authors illustrate the usefulness of these definitions by designing algorithms that satisfy them for various statistical tasks. The bulk of their technical work is to prove Lemma 3.1, Lemma 3.2, and Lemma 3.3, which are later used to establish various results. In particular, Lemma 3.1 is a more fine-grained version of a result from [VWDP+22], Lemma 3.2 is a randomized rounding scheme, that unlike the one proposed in [ILPS22], using a (relatively) small number of random bits, and Lemma 3.3 is a straightforward modification of Sperner's/KKM that has appeared in the literature.

First, they consider the problem of estimating the bias of $d$ independent coins from samples. They design a deterministic and efficiently computable algorithm that is $d+1$-list replicable and requires $O\left(\frac{d^3}{\epsilon^2}\cdot \log(d/\delta)\right)$ samples in total, where $\epsilon$ is the accuracy parameter and $\delta$ is the confidence parameter. At this point it is worth mentioning (even though not explicitly stated in the paper) that there exists a (trivial) $2^d$-list replicable algorithm whose sample complexity is $O\left(\frac{d}{\epsilon^2}\cdot \log(d/\delta) \right)$ -- the authors use this algorithm for a different task (more details on this later). For the same task, they design a $\log(d/\delta)$-certificate replicable algorithm with sample complexity $O(\frac{d^3}{\varepsilon^2\delta^2}$ in total. Notice that the sample complexity w.r.t. $\delta$ is $poly(1/\delta)$, whereas for the list replicable algorithm it is $poly(\log(1/\delta)$. They complement their upper bounds by showing that there is no $k$-list replicable algorithm for this task, for any $k < d+1$, and no $k$-certificate replicable algorithm for any $k < \log(d)$. This implies that the list complexity parameter is optimal and the certificate complexity parameter is optimal up to a factor of $\log(1/\delta)$, but it does not imply that the sample complexity of the algorithms is optimal.

Then, the authors consider the canonical learning task of binary classification in the PAC setting. They first show that any class $H$ that is learnable in the well-known SQ model by non-adaptive queries admits a $d+1$-list replicable algorithm whose sample complexity is $O\left(\frac{d^2}{\varepsilon^2} \log(d/\delta)\right)$. They further show that there is a $\log(d/\delta)$-certificate replicable algorithm for this task with sample complexity $O\left(\frac{d^2}{\delta^2\varepsilon^2} \log(d/\delta)\right)$. In the case of adaptive queries, the only list-replicable algorithm they present is the trivial one with $L=2^d$, but they design a $d\log(d/\delta)$-certificate replicable algorithm with sample complexity $O\left(\frac{d^3}{\delta^2\varepsilon^2} \log(d/\delta)\right).$

Finally, they consider the task of learning $d$-dimensional thresholds on $[0,1]^d$. They observe that if the marginal distribution on $X$ is unknown, there is no list-replicable algorithm for this task. Thus, they restrict the problem by fixing this distribution be uniform. They prove that this restriction helps circumvent the lower bound by showing that there exists a $d+1$-list replicable algorithm for this problem. Moreover, the list replicability parameter cannot be decreased.


[VWDP+22]: Jason Vander Woude, Peter Dixon, Aduri Pavan, Jamie Radcliffe, and N. V. Vinodchandran. Geometry of rounding.

[ILPS22]: Russell Impagliazzo, Rex Lei, Toniann Pitassi, and Jessica Sorrell. Reproducibility in learning.

**Strengths:**

This work is trying to address a very important problem in ML from a theoretical point of view. It is important to have algorithms with provable replicability guarantees and not just rely on empirical replicability of experimental findings.
Recently, there has been an interest from the TCS/theoretical ML community to design algorithms with mathematical replicability guarantees. The authors do a good job of placing their work in the literature.

I think the most exciting contributions of the paper are the replicability definitions they propose, which are novel and I find them quite natural and interesting. Moreover, even though the main technical challenges to establish Lemma 3.1, and Lemma 3.3 were already solved in prior work, the use of the rounding scheme of Lemma 3.1 based on secluded partitions in the replicability literature is novel and I believe that it can have further applications in this line of work. Similarly, using Sperner's/KKM lemma to establish lower bounds for list-replicable algorithms could be used to derive lower bounds for other replicability definitions that researchers might come up with in future work. The randomized rounding scheme that is proposed in Lemma 3.2 is a clever, though not very hard, tweak of the one in [ILPS22] that reduces the (potentially infinitely many) random bits that the naive application of [ILPS22] requires.

The paper is in general well-written and the proof sketches in the main body are easy to follow. The same thing is true for the appendix as well. That being said, I think there are some inaccuracies and some nuances that are not addressed appropriately, which I elaborate on in the Weaknesses/Questions section.

[ILPS22]: Russell Impagliazzo, Rex Lei, Toniann Pitassi, and Jessica Sorrell. Reproducibility in learning.

**Weaknesses:**

The biggest weakness of the paper is that the results are somewhat limited to simple problems. In particular, the results in Section 4 and Section 5.1 can be unified and stated as list/certificate-replicability results for SQ learning (see also my first question to the authors). The next result that the authors prove in the binary classification setting is restricted since it holds for a particular hypothesis class under a particular distribution on the feature space. One immediate extension the authors could consider would be to prove this result for any fixed distribution on $X$ and not just the uniform distribution. Please see below why I believe that this is true. Another obvious direction is to establish sample complexity lower bounds for the various algorithms you have developed.


Some inaccuracies/issues in the exposition:

-Line 140: As far as I can tell the notion of list-global stability [GKM21] does not coincide with the notion of list replicability. In particular, a list-globally stable learner outputs a list of hypotheses (not just a single one) and it is required that at least one of the elements of the list will be the canonical output, i.e., it will appear with high probability after every execution. On the other hand, a list replicable algorithm outputs a single element, but whp, this single elements belongs to a canonical list.

-Line 155-156: "Establishing the optimality of these measures". As far as I can tell, you didn't establish the optimality of the sample complexity.

-Observation 5.8: Not sure what it means for $k$ to be "constant". Even if we allow it to depend on parameters of the class or on $d$ the result is true.

-Proof of Observation 5.8: First of all, I think it is important to state the results of [ALMM19], [BLM20] that prove the claim (at least in the appendix) since many readers might not be familiar with these works. Moreover, to the best of my knowledge, the statement "any class $H$ has finite Littlestone dimension if and only if for some ``constant'' $k$ there is a $k$-list replicable algorithm for $H$" does not follow immediately from these two works. The direction that I can see as a direct result is that if $H$ has infinite Littlestone dimension then there does not exist a $k$-list replicable algorithm for $H$.
In particular [ALMM19] showed that every class that is approx-DP learnable has finite Littlestone dimension. Subsequently, [BLM20] showed that every Littlestone class has a globally stable learner with appropriate global stability parameters. Assume that H has infinite Littlestone dimension but admits a $k$-list replicable learner. Then, this gives rise to a globally stable learner with stability parameter at most something like $1/k$, which means that the class is approx-DP learnable, hence a contradiction. The converse direction can be obtained by Theorem 2 of [CMY23], however I do not see an immediate way to obtain it from [ALMM19],[BLM20]. Notice that a globally stable learner is not 1-list replicable.
Please let me know if I'm missing something.

-Line 368-369. This is not accurate. Be careful with the order of the quantifiers. [ALMM19] did not show that there exists a single "hard'' distribution $\mathcal{D}^\star$ for the class of one-dimensional thresholds. They showed that for any approx-DP learning algorithm $A$, there exists a hard distribution $\mathcal{D}_A$, i.e., the distribution depends on the algorithm. In fact, it is known that for any fixed $\mathcal{D}$, any VC class is approx-DP learnable.

-It is a bit confusing to refer to the distribution-dependent learning setting as the PAC learning setting, because most readers would assume that the PAC setting is distribution-free.

Typos and other comments that didn't affect the score/evaluation.

-Line 72 "multi-pseuodeterminism" -> multi-pseudodeterminism.

-Line 333, 342, 349, 547, 567, 589 "reproducibly" -> replicably.

-Line 100: "with at least" -> "with probability at least".

-Line 201: "randomize" -> randomized.

-Line 201: maybe you can restrict $\varepsilon$ to be at most one, outside this range it is trivial to satisfy the definition.

-Line 187-188: use singular form for the rounding scheme.

-It might be useful to state the formal definitions of replicability in section 1.1 (see also the next comment).

-Instead of having two separate definitions of an "accurate" list replicable and certificate replicable learner based on the statistical task (coins/PAC learning), it might be better to have one definition that captures just what list replicability/certificate replicability means and then specify what "correctness" means based on the underlying task, separately -- i.e., have definitions that say what it means for the learner to be list replicable/certificate replicable, have an (application specific) definition on what it means to be accurate and then the accurate list replicable/certificate replicable replicable algorithm is simply the composition of these definitions.

-It might be good to point out the difference between global stability [BLM20] and 1-list replicability. Global stability asks for one canonical output that appears with non-zero probability, whereas 1-list replicability asks for the canonical output to appear whp.


-Line 256-263: Maybe you can have it as a Remark/Observation.

-Line 272: You could have an one-line comment that this follows from the data-processing inequality. Also, it looks like the two subscripts of $\mathcal{D}$ are not aligned.

-Line 406: did you mean if $\vec{p} \in X$?

-Line 475: "are round" -> are rounded .

-Line 480: "almost" -> at most.

-Line 494: you mean $(d-1)$-dimensional face?

-Line 499-500: maybe it's good to define the closure formally (even though the reader can understand what it means).

-Line 698: set notation for $W$ missing.

-Line 704: "us" -> is.

-Line 705: "the" -> that.

[GKM21]: Badih Ghazi, Ravi Kumar, and Pasin Manurangsi. User-level differentially private learning via correlated sampling.

[ALMM19]: Noga Alon, Roi Livni, Maryanthe Malliaris, and Shay Moran. Private PAC learning implies finite littlestone dimension.

[BLM20]: Mark Bun, Roi Livni, and Shay Moran. An equivalence between private classification and online prediction.

[CMY23} Zachary Chase, Shay Moran, and Amir Yehudayoff. Replicability and stability in learning.

**Questions:**

1) I don't really understand what is the benefit of studying the coin estimation problem in Section 4 separately from the SQ learning problem in Section 5. In particular, as far as I can tell, all the proofs in Section 4 related to estimating the bias of coins hold verbatim for SQ estimation tasks. In fact, they can be phrased as general reduction: given a potentially non-replicable algorithm A that gives answers that are $\epsilon$-close to the true one in the $\ell_\infty$ norm, using the rounding schemes your propose one can get list\certificate replicable algorithms with the stated sample complexities/number of oracle calls to A. For instance, this can be used to replicably estimate potentially unbounded quantities that concentrate, such as averages of subgaussian random variables etc. Am I missing something?

2) As you have also pointed out in the manuscript, there is a trivial 2^d-list replicable algorithm for the coin estimation problem that needs O(d) samples (assume that $\epsilon,\delta$ is a constant) and your algorithm is $d$-list replicable but requires $O(d^3)$ samples. Are you aware of any $poly(d)$-list replicable algorithm that uses $o(d^3)$ samples?

3) Relatedly, do you believe that the $O(d^3)$ sample complexity is necessary to get the $d+1$-list replicability?

4) The dependence of the sample complexity of certificate replicability on the confidence $\delta$ is a bit unsatisfactory since it is $poly(1/\delta)$ and not $poly(log(1/\delta))$. I believe this is, to some extent, an artifact of the way you have defined certificate replicability. You can consider a two-parameter version of the definition that gives rise to more fine-grained sample complexity bounds. To be more precise, you can define an algorithm to be $(\ell, \rho)$-certificate replicable if $(1-\rho)$-fraction of the $\ell$-bit random strings $r$ have a canonical output $\mathcal{o}_r$. As in the non-replicable case, an algorithm is $(\epsilon,\delta)$-accurate if with probability $1-\delta$ it outputs an answer whose "error" is at most $\epsilon$. To define what it means to be $(\ell, \rho)$-certificate replicable and $(\epsilon, \delta)$-accurate you simply require that the algorithm satisfies both definitions. In your definition, you essentially treat the parameters $\delta, \rho$ in the same way. The advantage of the definition I suggest is that it will lead to sample complexities of the certificate replicable algorithms you propose that are of the form $poly(1/\epsilon, 1/\rho, \log(1/\delta)$. These are also the "correct" sample complexities that appear in this line of work (see, e.g., [ILPS22], [KKMV23]). Am I missing something?

5) In Line 211, shouldn't the randomness $r$ in the inner probability be fixed? In other words, shouldn't the inner probability be only wrt the random samples of the coins? (as you do in Def. 5.3)

6) In Theorem 5.9, why do you only consider the uniform distribution? I believe that the result should hold for any fixed distribution, but the way to construct the rounding scheme would depend on the distribution (i.e., it will not in general be the one you are proposing).

7) In the proof of Lemma 3.1, line 446, second equality, did you mean to write $\bar{B}_{\epsilon_0/\epsilon}^\infty(x/\epsilon)$? If I am not missing something, you can see that by picking $x=1$ for example, what you have currently written is not true.



[ILPS22]: Russell Impagliazzo, Rex Lei, Toniann Pitassi, and Jessica Sorrell. Reproducibility in learning.

[KKMV23]: Alkis Kalavasis, Amin Karbasi, Shay Moran, and Grigoris Velegkas. Statistical indistinguishability of learning algorithms.

**Limitations:**

The limitations are adequately addressed.

---

> ### Author Rebuttal · Authors · 2023-08-09
>
> Thank you for your detailed and very insightful comments, questions, and suggestions. We will take care of all typographic edits. We will first address the question and then address the weakness concerns.
>
> Q1: You are correct; a generic result as you suggested could be stated. We felt that the $d$-Coin Bias Estimation problem is a simple and natural problem that we could use to develop and introduce our techniques. Also, while the reduction from this problem to SQ learning gives upper bounds on other learning tasks, it does not give the lower bound which we establish in the paper. In our view the lower bound for the $d$-Coin Bias Estimation Problem (Theorem 4.7) is clean and fundamental.  The lower bound technique developed is generic and the lower bound for the $d$-Threshold builds on these techniques.
>
> Q2 & Q3: These are very interesting questions that we have been investigating. We can say that this is not possible if you use *rounding techniques* in the style of Lemma 3.1. Morally, rounding-based algorithms require $\Omega(d^3)$ samples up to polylog factors, even for ${\rm poly}(d)$-list replicability.
>
> Q4: Thank you for pointing this out. It is true that by using the suggested definition, we will obtain sample complexity of ${\rm poly}(1/\rho, 1/\varepsilon,\log (1/\delta))$. In fact, this is one of the choices we made while defining certificate complexity in order to reduce the number of parameters involved.
>
> Q5: Yes, $r$ should be fixed in the inner probability. Thank you for pointing this out. We will clarify this.
>
> Q6: We do not see how to extend the result to other distributions. Consider the case when $d = 1$. For this to work, we need the following: Given $\varepsilon$, divide the interval $[0, 1]$ into $1/\varepsilon$ intervals of equal probability mass. While such a division exists, we do not know how to compute it efficiently for arbitrarily fixed distributions in a uniform manner. An approach would observe samples from the distributions and try to get an approximate division. However, we do not know how to do this in a replicable manner. In general, the notion of $\varepsilon$-close threshold with respect to the uniform distribution corresponds to $\ell_{\infty}$ norm, and the secluded partitions can be used to do the rounding. When the distribution changes, the corresponding norm changes and it is not clear how to do the rounding.
>
> Q7: You are correct. We will fix this in lines 2 and 3. As you probably noted, this does not affect the correctness of the lemma.
>
> Address Weaknesses:  We really appreciate the detailed comments and suggestions. We will address them in the final version, especially Observation 5.8 and the discussion of results from ALMM19 and BLM20. Also as you point out, the list global stability, while similar, does not  coincide with list replicability. We will clarify this. In addition, taking your suggestion for unifying definitions,  and also based on two other reviewers suggestions, we have made an attempt to unify the definitions. These new  generalized definitions can be found in the global rebuttal.

---

> > ### Comment · Reviewer_U7Lz · 2023-08-10
> >
> > I would like to thank the authors for their detailed response that addresses all my questions.
> >
> > Q1: I agree that the $d$-Coin Bias Estimation problem is very clean and it could help the exposition. The choice of the authors makes more sense to me after reading their response.
> >
> > Q2 & Q3: This is very interesting. Is there an easy way to see that? If this is part of a separate ongoing work do not feel obliged to answer the question, I understand that sharing results of ongoing work in a public forum might not be the best idea.
> >
> > Q4: I understand that the modified definition I suggested increases the notational clutter, so it is up to the authors to decide how they want to state it. In any case, it might be helpful to add a short comment explaining that the dependence on the confidence parameter can be improved.
> >
> > Q6: I admit that I'm not aware of a computationally efficient approach to extend the result to other distributions, I hadn't thought of this obstacle. I understand the challenges the authors mentioned.
> >
> > Q7: I understand that it does not affect the correctness of the result (as reflected on my soundness score).
> >
> > Global rebuttal: The unified definitions look good to me.

---

> > > ### Author Response · Authors · 2023-08-11
> > > **Answer to Q2 & Q3**
> > >
> > > Thanks for your quick response.
> > >
> > > Answer to Q2 & Q3: It is an ongoing work. It is based on establishing tightness of parameters from the work of VWDPRV22 cited in the paper.
> > >
> > > Answer to Q4: We agree that it will be helpful and will add a comment  explaining that the dependence on the confidence parameter can be improved.

---

### Official Review · Reviewer_xJoj · 2023-07-04

**Soundness:** 4 excellent
**Presentation:** 2 fair
**Contribution:** 3 good
**Rating:** 6
**Confidence:** 5

**Summary:**

This paper considers two notions of complexity for stable algorithms: list complexity for list replicable algorithms and certificate complexity for pseudo-globally stable/replicable algorithms. They give matching upper and lower bounds for the list and certificate complexities of the d-coin bias estimation problem. They extend these results to obtain upper bounds on the complexities of adaptive and non-adaptive SQ algorithms and upper and lower bounds on the list complexity of learning d-dimensional thresholds under the uniform distribution.

**Strengths:**

This paper has a novel perspective on the study of replicable algorithms, by considering how much randomness is required to achieve replicability, and it continues the study of list bounds for list replicability. The rounding techniques and application of Sperner’s lemma used to prove their results are interesting, and I think the fact that they are able to obtain several results for fundamental learning problems from variations of these techniques speaks to their usefulness. I enjoyed reading the paper.


**Weaknesses:**

Though I enjoyed the paper overall, it was structured in a way that frequently made it difficult to understand or contextualize their results.
- Formal definitions of the complexity measures studied appear in the context of a particular problem, rather than being properties of general algorithms, which was a bit confusing.
-  The motivation for these complexity measures is not very clear to me.
- The explanation of how to simulate a statistical query algorithm making d adaptive queries is not specific enough to be enlightening. Either expanding it or omitting it would make the section more readable.
- As a reader, I didn’t gain much from the exposition on page 2 showing that perfectly replicability isn’t achievable, and I think it took space away from expanding on why list and certificate complexity are useful notions of non-replicability to minimize.

The techniques used seem interesting, but seem mostly to have been developed in VWDPRV22. The novel contribution of this paper appears to me to be the adaptation of these results to the problem of certificate and list complexity, and so I think the usefulness of these complexity measures needs much more attention in the paper.



**Questions:**

As already mentioned, I think these complexity definitions need more motivation. They seem interesting, but how does minimizing list and certificate complexity facilitate replicability? How does list replicability compare to pseudo-global stability?

**Limitations:**

Yes, the authors address broader societal impacts.

---

> ### Author Rebuttal · Authors · 2023-08-09
>
> Thank you very much for the review. We are glad that you enjoyed reading the paper. We will first address the question and then address the weakness concerns.
>
> Q Part 1: We will include additional motivation in the final version to the extent possible. As noted in the introduction, strong replicability is not achievable in the learning setting even for simple learning tasks. Consequently, a few natural and foundational questions are the following: Can we "measure" the degree of (non)-replicability of learning tasks? Are some learning tasks more non-replicable than others? The notions of list complexity and certificate complexity are two different ways to formalize the degree of (non)-replicability and answer these questions.
>
> For example, a $1$-list replicable algorithm is strongly replicable (i.e. with a high probability a single canonical answer is given over multiple runs). With a $2$-list replicable algorithm we see at most $2$ distinct answers, and with a $100$-list replicable algorithm we could see up to $100$ distinct answers. While a $1$-list replicable algorithm is the most desirable, when this is not possible, it is natural to desire a $2$-list replicable algorithm as opposed to, say a $100$-list replicable algorithm. For certificate replicability, once the certificate is fixed, the rest of the algorithm is strongly replicable; thus, an algorithm with zero certificate complexity is strongly replicable. A small certificate complexity implies that the learning task is closer to being strongly replicable than an algorithm with a higher certificate complexity.
>
> Q Part 2: The notion of pseudo-global stability is similar to the notion of certificate replicability, and as proved in the paper, a list replicable algorithm can be converted into a certificate replicable algorithm. Thus, a list replicable algorithm can be made pseudo-globally stable with a bound on the amount of randomness. However, this generic conversion is sub-optimal in terms of sample complexity.
>
> Addressing Weaknesses:
>
> W1: We agree that it is indeed possible to place all definitions in a single section and define the notions of list/certificate replicability as properties of algorithms rather than making problem specific. Two other reviewers suggested the same. Hence we have made an attempt to do so. The generalized definitions can be found in the global rebuttal.
>
> W2 & W4: Our purpose in this exposition was to demonstrate that strong replicability is not achievable even in simple settings, and we felt that this was necessary to justify why we study these weaker forms of replicability. This sets the stage and motivation for the rest of the paper. Since Reviewer Hncd also raised this concern, we will seriously consider moving the proof from the introduction to the technical part and give more motivation in the final version.
>
> W3: Thanks for the comment. We think it is an interesting result and we will give more details in the final version.

---

> > ### Comment · Area_Chair_UssN · 2023-08-18
> >
> > Dear authors,
> >
> > Your respondse has been noted.
> > The decision on your paper will be based on my discussion with the reviewers.
> > We will reach out to your should we require further clarifications.
> >
> > Regards,
> > AC

---

> > ### Comment · Reviewer_xJoj · 2023-08-20
> >
> > Thanks to the authors for their response! To clarify my original comment about motivations, I think the paper was clear that the studied complexity measures arise from the question of how to formalize the degree of non-replicability for an algorithm. I agree that this is a natural theoretical question arising from the impossibility of strong replicability for some tasks, but I'm curious about what, if any, insights these quantities give us into the tasks themselves beyond what is immediately implied by the definitions of list replicability and certificate complexity. If there are practical implications of lower/upper bounds on these measures, it would be great to highlight them.

---

> > > ### Author Response · Authors · 2023-08-21
> > >
> > > Thanks for the clarification. As you noted, the scope of this work is to formalize meaningful notions of replicability. We anticipate practical implications of the proposed replicability notions. For example, we envision that the notion of certificate replicability is applicable to optimize communication between parties in a distributed learning setting where the goal is to arrive at a consensus answer. However these ideas need to be explored further.

---

### Official Review · Reviewer_HcY8 · 2023-07-07

**Soundness:** 3 good
**Presentation:** 2 fair
**Contribution:** 3 good
**Rating:** 5
**Confidence:** 3

**Summary:**

The paper investigates replicable learning algorithms by proposing two notions of replicability, i.e. the list replicability and the certificate replicability. For the problem of estimating the biases of $d$ coins, it designs a $(d+1)$-list replicable algorithm and complements the result by showing that the list size is optimal. For PAC learning model, it shows that any hypothesis class that is learnable with $d$ non-adaptive SQs can be learned by a $(d+1)$-list replicable algorithm or a $\tilde{O}(\log d)$-certificate replicable algorithm.


**Strengths:**

Designing replicable algorithms is of interest to both the theoretical and applicational community. Empirically, it is important for the models to be reproducible under slight changes in the environment. From the theoretical side, the problem itself is interesting whether provable guarantees can be achieved for algorithmic replicability. The paper generalized the idea of replicability by proposing two notions, allowing the algorithms to be list-replicable or certificate replicable, which characterize the replicability of a broader class of problems from bias estimation of multiple coins to PAC learning thresholds.


**Weaknesses:**

Some parts of the presentation can be improved. For example, as major notions, the definition of list-replicability and certificate-replicability can be placed in Section 1. To now, the ideas seem to be implied from the definition of specific problems instead of standalone definitions, which makes the paper less clear.


**Questions:**

To what extent is the notion of list-replicability related to the notion of list PAC learning or list-decodable learning? Since the PAC learning model is considered in this paper, it would be interesting if these notions could be compared.


**Limitations:**

No concerns.

---

> ### Author Rebuttal · Authors · 2023-08-09
>
> Thank you very much for the review and pointers to related literature. We will first address the question and then address the weakness concern.
>
> Q: Thank you for pointing us to these notions of list PAC learning and list decodable PAC learning. We will discuss these notions and contrast them with the list/certificate replicability notions, in the final version. List PAC learning and list-decodable learning are generalized models of learning. For example, any learning task that is  PAC learnable is trivially list PAC learnable with a list size of 1. However, list/certificate replicable learning is an additional requirement that needs to be satisfied by a learner. Thus the notion of list and list-decodable PAC learning are different from the notions of list/certificate replicability.
>
> Addressing Weakness: We agree that it is indeed possible to place all definitions in a single section and define the notions of list/certificate replicability as properties of algorithms rather than making problem specific. Two other reviewers suggested the same. Hence we have made an attempt to do so. The generalized definitions can be found in the global rebuttal.

---

> > ### Comment · Reviewer_HcY8 · 2023-08-18
> > **Official Comments by Reviewer HcY8**
> >
> > Thank you very much for your response. I agree it will be helpful to include the comparison in the revision.

---

### Official Review · Reviewer_Hncd · 2023-07-31

**Soundness:** 3 good
**Presentation:** 2 fair
**Contribution:** 3 good
**Rating:** 6
**Confidence:** 2

**Summary:**

This work studies the theoretical property of replicability of machine learning algorithms. The paper's authors introduce two notions of list and certificate complexity used to quantify the replicability of such algorithms. They also establish replicable algorithms for the coin bias estimation problem and in the general setting of PAC learning, proving theoretical guarantees on the properties of these algorithms.

---
Post-rebuttal update: after reading the author response, I am increasing by confidence in the score of 6 by 1 point. I have not worked in the field of working theory, but the results are sensible to me and the rebuttal has addressed some of my questions.

**Strengths:**

* The work studies an important problem of having not only sample-efficient, but also replicable algorithms that would arrive to a narrow set of hypotheses after training on different samples from the data distribution.
* Although I am not an expert in learning theory, the theoretical analysis appears to be sound and the reasoning steps are explained in sufficient detail
* The paper covers the related work well, connecting the contributions of the authors to the existing results in a clear way

**Weaknesses:**

* I feel the work would strongly benefit from an improvement of its presentation. For example, the abstract is several paragraphs long. Furthermore, the introduction contains an example illustratory statement along with its full proof, which makes this section of the paper a bit dense.
* Some parts of the paper could also be improved by providing more insight into the motivation behind the introduced notions of replicability. It is not fully clear if the certificate complexity has any specific advantages over previous measures of replicability discussed in the CMY23 reference that would make it of interest to the community.

**Questions:**

* Could you provide some intuition into the certificate complexity, for example, into the meaning of the certificate $r$? The proof of Lemma 3.2 in the appendix contains a constructive description of what $r$ does and how it is used, but the work might be easier to understand if we had an example for any standard machine learning problem in a more practical setting (e.g., binary classification of real-valued vectors) that would use $r$ as an input.
* The introductory section provides useful insight into the motivation behind studing reproducibility, but (unless I am mistaken) it does not address two important real-world considerations. First, while machine learning algorithms observe samples from a distribution, in a supervised offline setting (which is the most popular one) they are trained on a fixed finite-sized sample from that distribution (usually called the training dataset). If the reproducing party trains on the same open dataset, that factor of randomness does not influence the resulting model. Second, all other hardware-based factors of randomness are usually also fixed by setting the random seed at the beginning of training. Would the analysis of replicability be still useful in such a case?

**Limitations:**

There is a discussion of limitations in the conclusion section, which was helpful in understanding the scope of the work.

---

> ### Author Rebuttal · Authors · 2023-08-09
>
> Thank you very much for the review. We will first address the questions and then briefly address the weakness concerns.
>
> Q1: One can think of the certificate as a "certificate of replicability." The definition captures the intuition that (1) most of the certificates should be good (2) once we fix a good certificate, the rest of the computation is strongly replicable.  In particular, after fixing a good certificate, the output of the algorithm doesn't depend on the samples drawn from the distribution. The proof of Lemma 3.2 indeed proves that a randomly chosen $r$ will be a good certificate. Proof of Theorem 4.6 uses Lemma 3.2 in the concrete setting of d-Coin Bias Estimation.
>
> Q2: In supervised learning, even though the data set is fixed, a typical algorithm will randomly select a subset of the data set for training, and the rest for validation and testing.  The randomly chosen training data set can be thought of as coming out of a distribution. The proposed framework of the paper applies to this scenario.  If we choose to view the data set and the random seed as fixed and there is no randomness involved, then the learning is a deterministic task. In this case, the notions of replicability introduced in this and related works are trivially achieved. The study of replicability in learning is an emerging research area and in this work, we focus on the PAC-style learning algorithms where the algorithm observes randomly chosen samples from a distribution.
>
> Addressing Weakness: The example given in the introduction shows that strong replicability is not possible even for very simple tasks. This sets the stage and motivation for the rest of the paper. Since Reviewer xjoj also raised this concern, we will seriously consider moving the proof from the introduction to the technical part and give more motivation in the final version.

---

> > ### Comment · Reviewer_Hncd · 2023-08-13
> > **Thank you for your response!**
> >
> > Thank you for the clarifications! They have addressed my questions (although I would prefer an example of a certificate in a more standard machine learning setting, as I mentioned in the initial review); to reflect that, I have increased my confidence in the score by 1 point. I do not feel I am qualified enough in this research area, therefore I can't champion for its acceptance with a higher score than I have currently given

---

> > > ### Author Response · Authors · 2023-08-21
> > >
> > > For a machine learning task, the certificate depends on the underlying learning task and the certificate replicable algorithm designed.  Typically certificates are randomly chosen, from a carefully chosen small universe,  as in the case of the certificate replicable algorithms designed in this paper. To illustrate, consider the 1-coin bias estimation problem, with delta = 1/16. The set of possible certificates are integers $0$ through $15$. For any given bias, a randomly chosen integer will be a good certificate of replicability (with high probability). However, we can not hope to apriori *deterministically* fix a certificate, as this will lead to strong replicability, which we have shown is not achievable.

---

### Author Rebuttal · Authors · 2023-08-09

As three of the reviewers suggested unifying replicability definitions for d-Coin Bias Estimation and PAC learning, we have made an attempt to do so. We present it below. Let us know your thoughts on whether this is better as opposed to what is in the submission.

A learning problem is a family $\mathcal{D}$ of distributions over a domain $X$ along with a set $H$ (representing hypotheses) along with an error function ${err}: \mathcal{D} \times H \rightarrow [0,\infty)$. The goal is to learn a hypothesis from $H$ based on some $D\in\mathcal{D}$, and we focus on the realizable case.

A learning algorithm $A$ has the following inputs: (i) $m$ independent samples from a distribution $D\in\mathcal{D}$ and (ii) $\varepsilon\in(0,\infty)$ and (iii) $\delta\in(0,1]$. It may also receive additional inputs.

**List-Replicability**

Let $k \in\mathbb{N}$, $\epsilon\in(0,\infty)$, and $\delta\in[0,1]$.
A learning algorithm $A$ is called $(k, \varepsilon, \delta)$-list replicable if the following holds: There exists $m \in \mathbb{N}$ such that for every $D\in\mathcal{D}$, there exists a list $L\subseteq H$ of size at most $k$
such that (i) for all $h \in L$, $err(D, h) \leq \varepsilon$, and (ii)

$$\Pr_{s \sim D^m} [A(s, \varepsilon, \delta) \in L] \geq 1 -\delta. $$

For $k\in\mathbb{N}$, we call $A$ $k$-list replicable if for all $\epsilon\in(0,\infty)$ and $\delta\in(0,1]$, $A$ is $(k,\epsilon,\delta)$-list replicable.

**Certificate Replicability**

Let $\ell\in\mathbb{N}$, $\epsilon\in(0,\infty)$, and $\delta\in[0,1]$. A learning algorithm $A$ is called $(\ell,\epsilon,\delta)$-certificate replicable if the following holds: There exists $m\in\mathbb{N}$ such that for every $D\in\mathcal{D}$ there exists $h:\\{ 0,1\\}^\ell\to H$
such that

$$
    \Pr_{r \in \\{0,1\\}^{\ell}}\Big[
        \Pr_{s \sim D^m}\big[
            A(s, \varepsilon, \delta, r) = h(r) \text{ and } err(D,h(r))\leq\epsilon
        \big] \geq 1-\delta
    \Big] \geq 1- \delta.
$$

**$d$-Coin Bias Estimation Problem**
    For the $d$-Coin Bias Estimation Problem, we have the following. $X=\\{Head,Tail\\}^d$ which is the set of representations of all possibilities of flipping of each of $d$-many coins. $H=[0,1]^d$ which is the set of all $d$-tuples representing biases of a collection of $d$-many coins. $\mathcal{D}$ is the set of all $d$-fold products of Bernoulli distributions over the symbols $\\{Head,Tail\\}$, so each distribution $D\in\mathcal{D}$ can be represented by a vector $v_{D} = \langle b_1, \dots, b_d\rangle \in [0,1]^d$ which is the vector of probability parameters of the $d$ Bernoulli distributions. Lastly, the error function is defined as $err(D,h)=\Vert{v_D-h}\Vert_\infty$.

**PAC Learning Model**
    For the PAC learning model we have a domain $X'$, a hypothesis class $H$ which is some collection of functions $X'\to\\{0,1\\}$.
    For any distribution $D$ over $X'$ and any  $f \in H$, let $D_f$ denote the distribution over $X'\times\\{0,1\\}$ obtained by sampling $x\sim D$ and outputting $\langle x, f(x)\rangle$. We define the following learning problem in our context: $X=X'\times\\{0,1\\}$, $H=H$, $\mathcal{D}=\\{D_f:f\in H\text{ and }D\text{ a distribution over }X'\\}$, and $err(D_f, h)=\Pr_{x \sim \mathcal{D}'} [f(x) \neq h(x)]$.

---

### Decision · Program_Chairs · 2023-09-21

**Decision:**

Accept (spotlight)

**Comment:**

This paper studies notions of replicability in learning and relies on interesting links with topology to study them. The reviewers agreed that this paper should be accepted. We decided to accept it as a spotlight.